# Complementary mesoscale dynamics of spectrin and acto-myosin shape membrane territories during mechanoresponse

Andrea Ghisleni[1,4 ✉], Camilla Galli[1,4], Pascale Monzo[1], Flora Ascione[1], Marc-Antoine Fardin[2], Giorgio Scita [1,3], Qingsen Li[1], Paolo Maiuri [1] & Nils C. Gauthier[1 ✉]

The spectrin-based membrane skeleton is a major component of the cell cortex. While expressed by all metazoans, its dynamic interactions with the other cortex components, including the plasma membrane or the acto-myosin cytoskeleton, are poorly understood. Here, we investigate how spectrin re-organizes spatially and dynamically under the membrane during changes in cell mechanics. We find spectrin and acto-myosin to be spatially distinct but cooperating during mechanical challenges, such as cell adhesion and contraction, or compression, stretch and osmolarity fluctuations, creating a cohesive cortex supporting the plasma membrane. Actin territories control protrusions and contractile structures while spectrin territories concentrate in retractile zones and low-actin density/inter-contractile regions, acting as a fence that organize membrane trafficking events. We unveil here the existence of a dynamic interplay between acto-myosin and spectrin necessary to support a mesoscale organization of the lipid bilayer into spatially-confined cortical territories during cell mechanoresponse.

[1] Institute FIRC of Molecular Oncology (IFOM), Via Adamello 16, 20139 Milan, Italy. [2] Institut Jacques Monod, Centre National de la Recherche Scientifique UMR 7592 and Université Paris Diderot, 75013 Paris, France. [3] University of Milan, Department of Oncology and Hemato-Oncology, Via Santa Sofia 9/1, 20122 Milan, Italy. [4] These authors contributed equally: Andrea Ghisleni, Camilla Galli. ✉email: andrea.ghisleni@ifom.eu; nils.gauthier@ifom.eu

Eukaryotic cells have developed several mechanisms to control their shape, sense their surroundings, and adapt to external cues. While a lot of efforts have been devoted to the study of the actomyosin and microtubule cytoskeletons, our understanding of cytoskeletal scaffolds directly connected to the plasma membrane (PM) lags behind. These systems are expected to play crucial roles in many cellular mechanoadaptive processes by shaping PM topology in association with the underlying cell cortex. Such processes have been investigated at nanoscale resolution through electron microscopy or advanced light microscopy, and their nanoscale dynamics has just begun to be revealed by a handful of high-end microscopy techniques[1–3]. However, distinct mesoscale domains of nonpolarized PM/cortical components (1–10 $\mu m^2$ in size) lack details on the topological and dynamic reorganization during changes in cell shape and mechanics.

An important element of the PM-cortex composite material is spectrin. This protein expressed by all metazoans is able to assemble into a nonpolarized meshwork connected to the PM, the actin cytoskeleton, and its associated proteins[4,5]. In mammals, 7 different spectrin isogenes encode for 2 α and 5 β subunits, which can be alternatively spliced into different isoforms. Among them, αII- and βII-spectrins are the most expressed in solid tissues[6,7], whereas αI and βI expression is restricted to circulating erythrocytes. At the protein level, spectrin exists as an elongated head-to-tail α/β heterodimer juxtaposed to a homologous molecule via tetramerization domains. This spectrin tetramer retains at both ends two ABDs harbored by the two N termini of β-spectrins, while several PM-binding domains are present along with both α and β subunits. These binding domains are the key elements for anchoring the spectrin-based membrane skeleton to the actin cytoskeleton and the inner leaflet of the lipid bilayer[8].

The spectrin skeleton has been implicated in many processes, including the stability and organization of PM, signal transduction processes, and membrane trafficking via endo- and exocytic pathways[9]. In accordance with its broad range of physiological functions, αII- and βII-spectrin genes have been found to be essential in embryonic development[10], and are also involved in many pathological conditions[11].

Despite this wealth of knowledge, our understanding of spectrin macromolecular organization is limited to the study of ex vivo erythrocytes and neurons, where it forms a triangle-like lattice and a repetitive barrel-like array interspaced by actin nodes, respectively[12–17]. Interestingly, erythrocytes do not possess actin stress fibers at their cortex. They can only polymerize short actin filaments made of 13–15 G-actin monomers ($\approx$33 ± 5 nm in length) that specifically serve to cross-link multiple spectrin rods, which act as the exclusive PM-supportive scaffold[18,19]. Several attempts to describe the spectrin meshwork organization at high resolution have been reported by different electron and fluorescence light microscopy techniques. The reported lengths of the spectrin tetramer range from 50 to 200 nm, depending on erythroid or neuronal lineage derivation and sample preparation protocols[13,14,20]. To reconcile these disparate observations, a working model has been proposed whereby the spectrin mesh can stretch and relax at maximum contour length upon mechanical perturbation to preserve PM integrity and to maintain cell shape[5,21]. The elasticity of the meshwork is ensured by the intrinsic flexibility of the so-called spectrin repeats[22,23].

Recent studies in red blood cells show that spectrin is critical in preserving cell shape by working in conjunction with myosin-dependent contractility[24]. Whereas, in *C. elegans* neurons, spectrin protects axons from deformation by keeping them under constant tension in conjunction with the microtubules[25]. In the same model organism, spectrin and actin polymerization deficiencies have been shown to impair body axis elongation, supporting a cooperative mechanoprotective mechanism of the two cytoskeletons at the tissue scale[26]. βII-spectrin has also been involved in the maintenance of epithelial cell–cell contact through microtubule-dependent processes, and its dynamics was shown to inversely correlate with endocytic capacities[9]. A mechanoresponsive role during myoblast fusion in muscle development has recently been proposed for the αII/βV-spectrin dimer[27]. This developmental process is conserved among different species (e.g., drosophila and mammalian cells), lending support to the possibility that the more ubiquitously expressed αII/βII-spectrin plays a more general and widespread role in mechanoresponsive processes.

Here, we use a wide range of mechanobiology techniques to comprehensively analyze βII-spectrin behavior during cell mechanoresponse. We find that spectrin is a major dynamic component for shaping the mesoscale-topological organization of the cell cortex upon mechanical stimuli. Specifically, spectrin complements cortical actin distribution and dynamics, but they cooperate during mechanical challenges. We also unveil a fundamental role for myosin-driven contractility in the regulation of spectrin dynamics, and how the orchestrated interplay between spectrin and PM might complement the actin-driven pickets and fencing mechanism in regulating membrane-trafficking events, such as clathrin-mediated endocytosis (CME).

## Results

**Spectrin and actin define complementary PM territories**. The spectrin-based membrane skeleton has been shown to adopt different configurations in erythrocytes and neuronal axons[13,19], while the organization in other cell types is far less accurately depicted. To fill this gap, we examined the spectrin–actin supramolecular organization in a variety of mammalian cells. We focused on βII-spectrin, the most abundant among the β subunits in nucleated cells[28]. In mouse embryonic fibroblasts (MEFs), the two endogenous subunits (αII and βII) showed, as expected, a perfect colocalization by total internal reflection microscopy (TIRFM) (Supplementary Fig. 1A). On the contrary, endogenous βII-spectrin and actin displayed a remarkable complementary pattern, which was particularly prominent along the actin stress fibers that were devoid of βII-spectrin (Fig. 1a–c). This peculiar arrangement was conserved in many other cell types, primary or immortalized, of human and murine origin, derived from normal or pathological tissues at whole cell (Supplementary Fig. 1D), but particularly adjacent to the basal PM using TIRFM (Supplementary Fig. 1D and zooms in Supplementary Fig. 2). Specifically, βII-spectrin formed a mesh-like pattern that filled the gaps between long actin cables, and was completely excluded from actin-rich leading-edge structures such as lamellipodia and filopodia (Supplementary Fig. 2). Overall, we identified four subcellular regions of spectrin–actin complementarity in all cell lines tested: leading-edge, stress-fiber-enriched cortex, actin- or spectrin-rich membrane curvatures (Fig. 1a and Supplementary Fig. 2). Interestingly, actin-depleted membrane curvatures were highly enriched in βII-spectrin and vice versa, suggesting that the two scaffolds might aid in shaping negatively curved PM regions. It is worth noticing that in cortical regions prominently enriched in spectrin-based membrane skeleton, a faint actin staining could still be observed in highly overcontrasted images (Supplementary Fig. 1A).

The complementary pattern observed between spectrin and actin in cells seeded on a continuous adhesive substrate may not reflect the cortical organization of nonadhesive zones, such as on the apical part of the cell. To overcome this limitation, we applied microcontact printing techniques to create fibronectin-coated

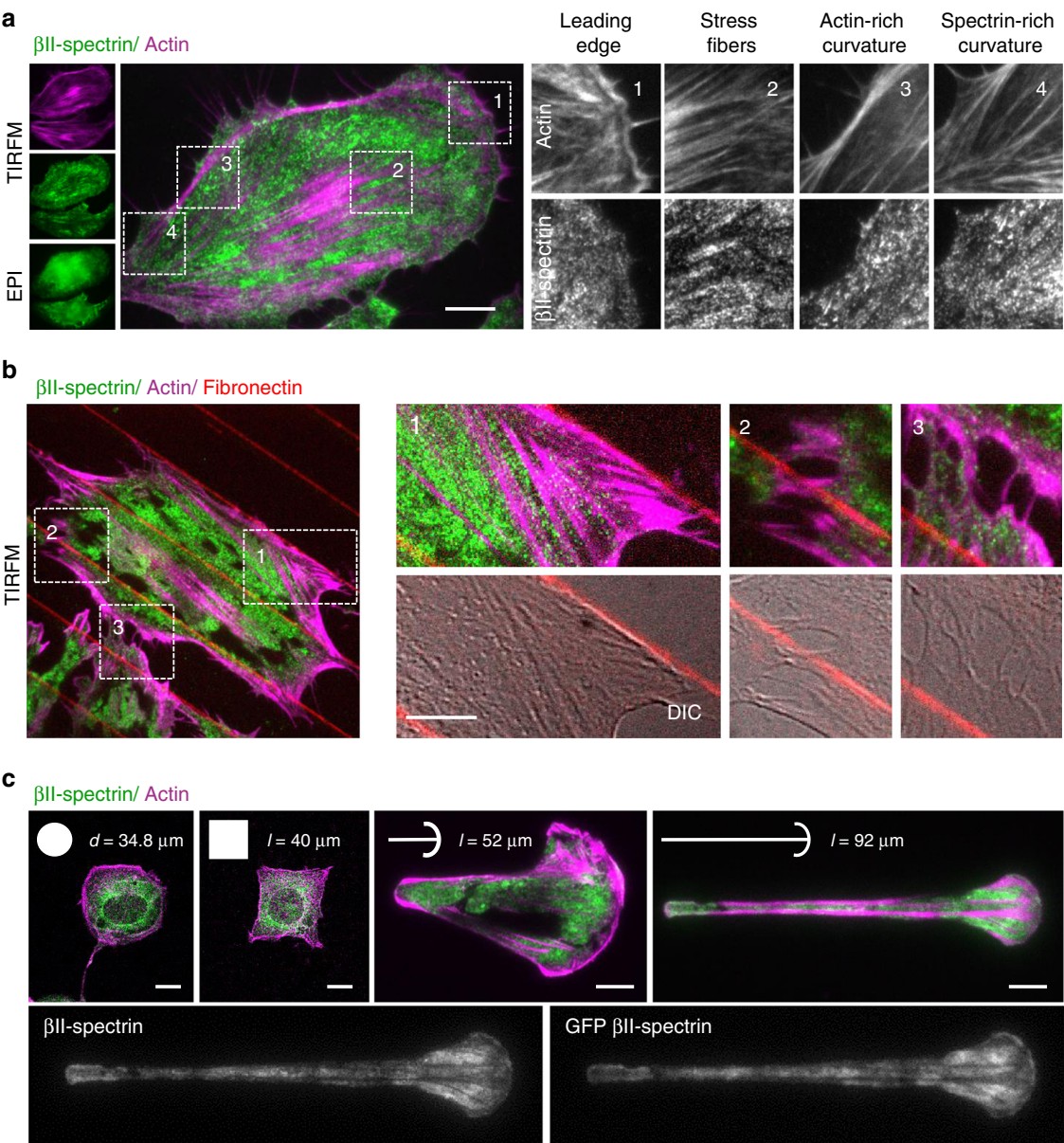

**Fig. 1 βII-spectrin and actin define distinct and complementary plasma membrane territories. a** MEFs immunostained for endogenous βII-spectrin (green) and F actin (magenta), observed by simultaneous TIRFM and EPI-fluorescence microscopy (scale bar: 10 μm). Four different cell zones are highlighted by dashed boxes (1–4), displaying regions by distinct morphological features. **b** MEFs seeded between adhesive fibronectin lines (red) and nonadhesive substrate (black) are visualized by TIRFM (endogenous βII-spectrin in green and F actin in magenta, scale bar: 5 μm). Three different zones are highlighted by the dashed boxes: (1–2) cell adhesions, (3) cell–cell contact. **c** Different geometries have been imposed to cells: circle and square (confocal), short and long crossbow (TIRFM). Immunostaining for endogenous βII-spectrin (green) and F-actin (magenta) is shown. Cells on the longer crossbow are transfected for GFP-βII-spectrin, immunostained for both endogenous and GFP-transfected proteins (scale bar: 10 μm, dimensions of the fibronectin-patterned geometries are also reported). All images are representative of at least 5 cells for each condition, immunolabelled in $n = 2$ or more independent experiments.

patterns separated by passivated nonadhesive surface and use the resolving power of TIRFM over nonadherent membrane patches (i.e., freestanding cortex-mimicry zones, Fig. 1c). Also, under these conditions, βII-spectrin and actin displayed a complementary pattern at stress-fiber-enriched cortex and membrane curvatures (Fig. 1c, crossbows in particular). Furthermore, by imposing different shapes to the cells from nonpolarized (circle, stress-fiber-poor) to polarized ones (long crossbow, stress-fiber-rich), we confirmed the exclusion of spectrin from actin-rich zones. Finally, this distinctive distribution of spectrin and actin was also observed in fixed cells using fluorescently tagged GFP-βII-spectrin (Fig. 1c).

**Spectrin dynamics during cell-driven mechanoresponse.** Fibroblast spreading can be considered as a stereotypical model to study de novo cytoskeletal assembly and cell-driven mechanoresponse (Fig. 2a)[29,30]. Naive suspended cells rapidly spread over matrix-coated substrates through a multiphasic process characterized by the initial cell attachment (P0) followed by the isotropic expansion of the cell area by Arp2/3-dependent actin polymerization (P1). After a short transition driven by a change in the PM tension (T), the activation of myosin contractility and membrane trafficking occurs (P2). This phase is characterized by a slower spreading rate, the maturation of focal adhesions, and the formation of stress fibers[31–33].

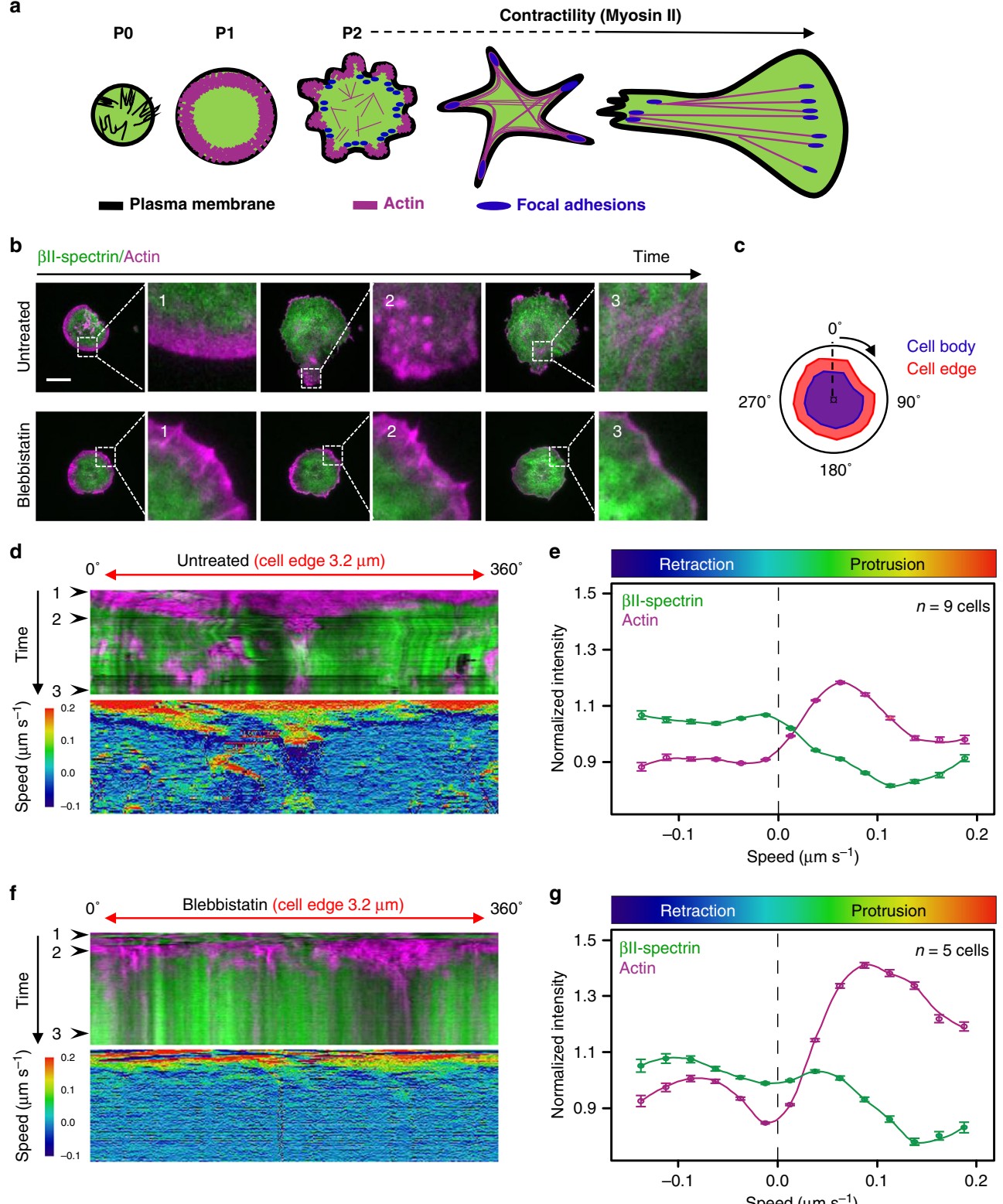

To investigate βII-spectrin recruitment to the PM during spreading, we fixed MEFs at different time points after seeding (within 5–20 min). We found a linear correlation between the amount of endogenous βII-spectrin and the projected cell area in the TIRF plane, likely reflecting the ability of βII-spectrin to associate constantly with the PM (Supplementary Fig. 3C, D, Supplementary Table 1). Actin signal, instead, deviated more significantly from linearity as a result of a more complex and dynamic behavior during the different spreading phases, such as the transition from a lamellipodia-driven in P1 to a stress-fiber-driven behavior during polarization. We also confirmed the βII-spectrin exclusion from actin-rich protruding edges (Supplementary Fig. 3A, B), in agreement with the observations at the leading edge of polarized cells (Fig. 1). However, the apparently constant spectrin/PM ratio measured at the whole-cell scale was more heterogeneous at subcellular level and

**Fig. 2 βII-spectrin and actin dynamics during spreading. a** Schematic representation of the different phases occurring during fibroblasts spreading on fibronectin-coated surfaces. The morphological changes in terms of cell shape, actin cytoskeleton (magenta), focal adhesion formation (blue), and PM organization (black) are drawn. **b** Cells untreated and treated with blebbistatin are visualized by live TIRFM, and representative images at relevant time points are shown (green: GFP-βII-spectrin, magenta: RFP-actin, scale bar: 20 μm, $n = 9$ cells (untreated) and $n = 5$ cells (blebbistatin) from independent experiments were analyzed). Peculiar mechanisms are highlighted by white dashed boxes and zoomed-in panels 1–3. In 1 is shown a typical noncontractile phase (P1), while the contractile phase (P2) is shown in 2; in the absence of myosin-II-dependent contractility (blebbistatin) cell spreading is stalled. In 3, coalescent actin nodes contribute to the maturation of the actin cytoskeleton, while blebbistatin treatment impairs these dynamic processes. **c** Schematic representation of the radial segmentation of the cell edge (red, 3.2-μm thickness) and cell body (blue) performed during the time lapse. **d–f** Radial kymograph analysis of cell-edge behavior of MEFs untreated (**d**) and treated with blebbistatin (**f**) is presented. The upper kymographs represent the integrated intensities of the two proteins (1–3) black arrowheads indicate the specific frames highlighted in (**b**), while the bottom kymographs display the edge speed related to the cell centroid. In (**e**) and (**g**), fluctuations from the mean signal intensities are plotted (actin: magenta and βII spectrin: green) in function of speed (untreated: $n = 9$ cells, blebbistatin: $n = 5$ cells, data are presented as mean values ± SEM).

evolved during spreading. In live cells, the analysis of the dynamics of GFP-βII-spectrin and RFP-actin throughout all the different phases of spreading confirmed the dynamic complementarity of the two skeletons. In particular, actin was invariably associated with protrusive processes that promoted cell area growth, while βII-spectrin displayed a passive-like behavior and was enriched in nonprotrusive PM regions (Fig. 2b, Supplementary Movie 1). Radial kymographs were generated to independently correlate βII-spectrin and actin fluorescence intensities with the local edge speed (see Methods), where protrusions (positive speed) and retractions (negative speed) occurred over time (Fig. 2c–g). βII-spectrin and actin intensities displayed opposite trends. Actin intensity peaked in protruding lamellipodia (≈0.08–0.12 μm sec$^{-1}$) as previously observed[32]; instead, it decreased in regions corresponding to highly positive speeds (>0.15 μm sec$^{-1}$), consistent with the possibility that actin becomes diluted in fast-protruding lamellipodia[34,35], and from null to negative speeds (Fig. 2d, e). βII-spectrin intensity displayed an opposite trend and was enriched at retracting zones characterized by negative speed independently from myosin-II contractility (see blebbistatin treatments, Fig. 2f, g). Peculiar edge-collapsing events during the contractile phase (P2), were marked by a sudden increase in βII-spectrin intensity (Supplementary Fig. 5A, B, Supplementary Movie 5). Global inhibition of contractility retained the opposite dynamics of βII-spectrin and actin (Fig. 2f, g). When spreading was performed on fibronectin-coated microprinted lines, cells were forced to form negative curvatures over the nonadhesive substrate in between adhesive stripes. Similarly, we observed retracting spectrin-rich curvatures to form upon actin withdrawal after initial lamellipodia-driven protrusion (Fig. 3a, b, Supplementary Movie 2), as well as during cell polarization (Fig. 3c, d, Supplementary Movie 2). Actin-rich curvatures were instead mainly observed during later spreading phases and were associated with protrusive cell portions (Fig. 3e, f), as we previously reported[36]. Altogether these quantitative dynamic observations provide support to a model whereby actin/spectrin are mutually exclusive both spatially and temporally along with the cell leading edge during fast remodeling events, and suggest the involvement of the spectrin mesh during cellular retraction.

We next focused our attention to the βII-spectrin dynamics under the cell body during spreading (Fig. 4a–c, Supplementary Movie 3). Fixed and live TIRFM analysis showed that the spectrin meshwork is progressively deployed and laid down from the back of the leading edge during P1 (Fig. 2b and Supplementary Movie 1), while apparent slight condensation in the lamella region was observed (Supplementary Movie 1 and 2). Consistently, thin confocal section analysis of the dorsal cortex in P1 revealed a homogeneous intermingled acto-spectrin meshwork behind the lamellipodia (Supplementary Fig. 3A, B). This indicated that both the noncontractile dorsal cortex and the

ventral one displayed similar organization. During P2, the spectrin meshwork under the cell body underwent a drastic remodeling corresponding to the increased actomyosin dynamics (Supplementary Movies 1–3). Actin nodes were formed in this specific spreading phase, priming stress-fiber maturation by condensation (Supplementary Movie 3, Fig. 4a, b). Remarkably, the spectrin mesh appeared to move in coordination with these expanding and condensing nodes, albeit not showing colocalization with those actin structures at TIRFM resolution. Myosin-II inhibition prevented such remodeling events without affecting the mutually exclusive actin/spectrin distribution at the cell edges, or the formation of poorly mobile actin nodes in spectrin-depleted zones (Fig. 2f, g, Supplementary Movie 1 and 3). Cross-correlation PIV analysis of actin and βII-spectrin flows highlighted areas of coordinated motion in terms of magnitude and directionality. This correlation landscape was analyzed during P2 in cells untreated and treated with blebbistatin, highlighting a significant decrease in size for areas of correlated motion (Fig. 4c, yellow zones) upon contractility inhibition (Fig. 4d). Thus, spectrin and actomyosin define membrane territories (up to 100 μm$^2$) by moving in a coordinated manner, clearly highlighting that the supramolecular mesh-like organization of spectrin is dynamically cross-organized by actomyosin remodeling.

The critical role of the actomyosin cytoskeleton in βII-spectrin dynamic organization was confirmed by monitoring protein flows after latrunculin A and blebbistatin washout experiments (Fig. 4e, f, Supplementary Movie 4). Consistent with the physiological observation in spreading cells, βII-spectrin expanded and redistributed upon actomyosin stress-fiber dissociation, and further augmented at cell leading edges upon cell retraction. During the drug washout phase, actomyosin nodes drove local βII-spectrin coalescence as cells restored their cytoskeletal architecture (Fig. 4e, f, Supplementary Fig. 3F and Supplementary Movie 4). The formation of actin nodes in spectrin-less zones was also confirmed by monitoring the distribution of endogenous proteins after latrunculin A washout in freestanding cortex-mimicry zones between patterned fibronectin lines (Supplementary Fig. 3F). These results further indicate that a similar coordinated organization of the spectrin and actin meshworks occurs in the nonadhesive cell cortex.

We conclude that the spectrin-based membrane skeleton is an almost continuous meshwork tightly associated with the PM, while its local density can largely fluctuate upon changes in cell geometry, dynamics, and mechanics. Spectrin locally condenses during events characterized by low actin-PM interaction, such as during membrane retraction at cell edges or the remodeling of cortical actomyosin nodes that lead to the formation of actin stress fibers, ultimately defining spectrin-rich territories.

**Cortex topology and contractility impact spectrin mobility.** Different mesoscale dynamic properties of βII-spectrin were

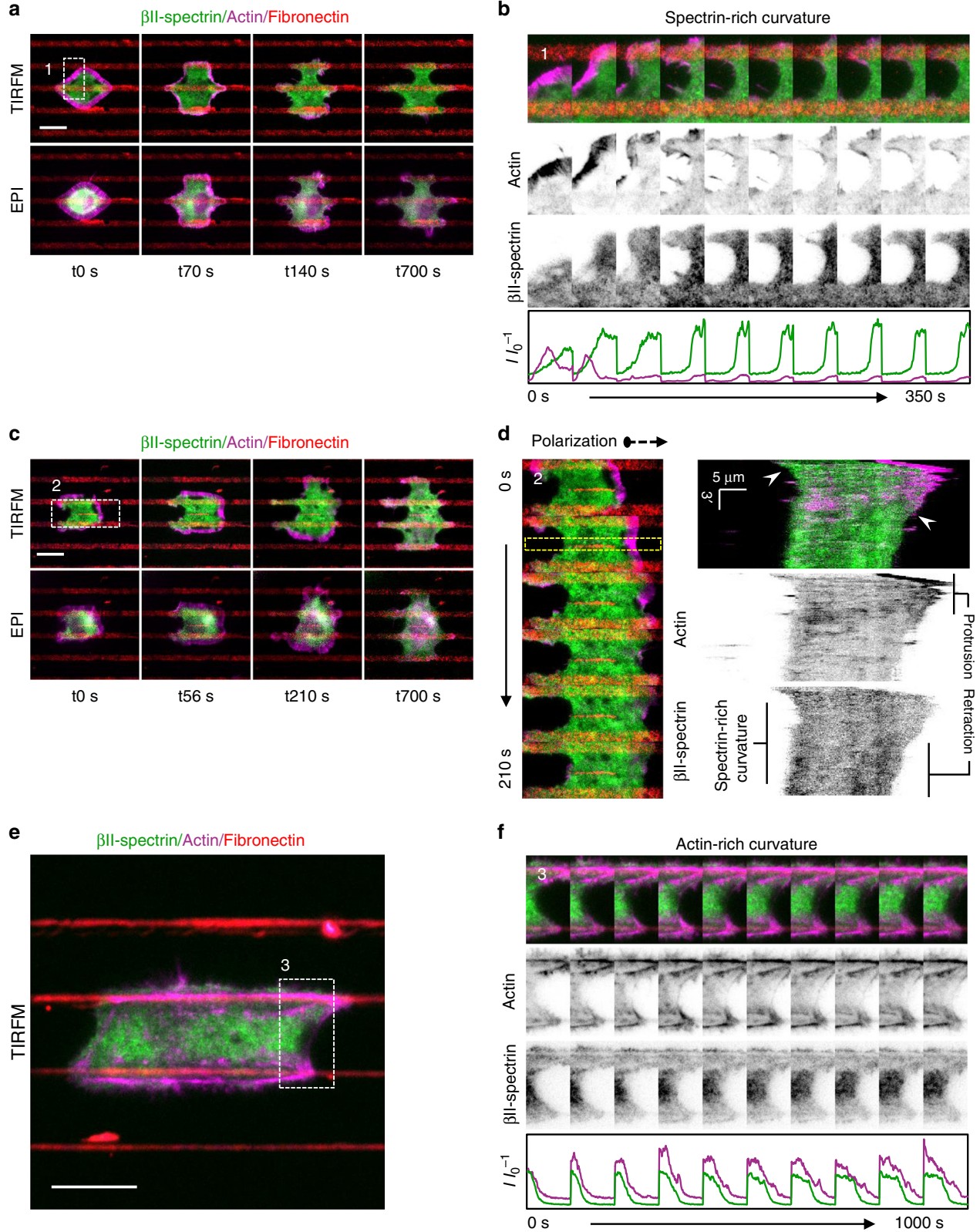

studied by fluorescence recovery after photobleaching (FRAP) at different cell locations. In particular, we focused on cortical regions defined by different topological organization described earlier: spectrin cortex, stress-fiber-rich zones, actin-rich, and spectrin-rich membrane curvatures (Fig. 5a). In MEFs co-transfected for GFP-βII-spectrin and RFP-Actin, the actin signal marked the four different locations. Cortical βII-spectrin was characterized by low-actin content and displayed higher mobility compared to the other zones fenced between actin stress fibers or negative curvatures (Fig. 5b–d). Half-time

**Fig. 3 Fibroblasts spreading between adhesive lines highlight βII-spectrin and actin complementarity. a** Representative images of MEFs immunostained for GFP-βII-spectrin (green), RFP-actin (magenta), and fibronectin (red), visualized by simultaneous live TIRF and EPI-fluorescence microscopy (scale bar: 20 μm). Time lapses lasted for 30 min; four representative frames are presented to describe the progression of the spreading. **b** Zooms related to the white dashed box presented in A across parallel fibronectin-coated lines and the passivated nonadhesive surface (in black). Normalized intensity profiles for both channels are presented to highlight the non-colocalization between βII-spectrin and actin during lamellipodia protrusion (first two frames) and subsequent retraction characterized by βII-spectrin-rich negative curvature. **c** Representative images of micropattern-induced polarization during spreading (scale bar: 20 μm). The zooms related to the white dashed box (2) and the morphological changes during the time lapse are presented in (**d**). Kymograph across the cell over the nonadhesive surface is drawn (yellow dashed box); white arrowheads highlight the beginning of βII-spectrin accumulation in the curvature regions. **e** Representative image of late-spreading phases, where the cell presents actin bridges over the nonadhesive substrate (GFP-βII-spectrin: green, RFP-actin: magenta, and fibronectin: red, visualized by live TIRFM, scale bar: 20 μm). **f** GFP-βII-spectrin accumulation at the nonadherent portion of the cell, remarkably followed the forward movement of the leading actin bridge. Images are representative of $n = 3$ independent experiments.

recovery rates were overall highly variable, potentially influenced by many protein-binding events independent from actin that could not be discerned here.

Since mesoscale spectrin condensation was frequently observed, we next sought to determine cytoskeletal mechanisms accountable for this reorganization. Pharmacological perturbation during live TIRFM was performed, and the mean GFP-βII-spectrin fluorescent intensity over the projected cell area was measured over 30 min (Supplementary Fig. 4A, B). We validated our approach using the oxidative cross-linker diamide, a drug that targets sulfhydryl groups and induces spectrin recoiling into ring-like structures[37,38]; βII-spectrin intensity increased significantly already after 5 min. The blockage of either de novo actin polymerization (by latrunculin A) or myosin-II contractility (by blebbistatin), did not alter whole-cell spectrin intensities, but only induced local fluctuations (Supplementary Fig. 4C). Spectrin condensation can be driven by actomyosin contractility at mesoscale level (as shown in Fig. 3), while the overall behavior at the whole cell must rely on alternative mechanisms. On the other hand, actin filament stabilization or microtubule (MT) depolymerization by jasplakinolide and nocodazole perturbations, displayed an overall ≈20% fluorescence increase after 5 min of treatment. As already proposed for βII-spectrin and βV-spectrin dynamics at cell–cell junctions[9], our results support that MT can control, at least in part, the spectrin-based membrane skeleton organization. However, given the broad effects of MT depolymerization on membrane trafficking, we cannot exclude that nocodazole treatment, known to block exocytosis and not endocytosis[30], may reduce the overall PM area driving the apparent spectrin condensation.

We elucidated these intricate observations by studying βII-spectrin turnover by FRAP analysis upon the same pharmacological treatments. A dual-FRAP assay on high-mobility cortical spectrin zones was performed on the same cell before and 5 min after drug administration to avoid secondary effects driven by long-term cytoskeletal perturbation (Supplementary Fig. 4A, B)[39,40]. Highly variable half-time turnover estimates were observed as previously documented (Fig. 5c), which precluded any interpretation. On contrary, consistent reduction was observed at the level of mobile fractions after treatment with blebbistatin and jasplakinolide (Fig. 5e, f, Supplementary Table 2), while latrunculin A and nocodazole failed to affect molecular turnover in the temporal window of this experiment. As expected, diamide-treated cells showed severely reduced βII-spectrin mobility. Overall, our results show that βII-spectrin molecular turnover relies directly on myosin-II-dependent contractility, and pharmacological stabilization of F-actin structures is reflected also on spectrin dynamics. At the whole-cell level, spectrin behavior is instead strongly dependent on cell architecture as expected from a shape-determinant meshwork.

**Actin binding coordinates spectrin dynamics**. Since the spectrin meshwork dynamics depended on actomyosin contractility as well as membrane topology, we discerned the contribution of spectrin domains that bind to actin or ankyrins/PM by deletion mutants (Fig. 6a). The ABD is present only in the β subunit that also harbors at least 3 PM-anchoring points[6]. We generated mutants of βII- spectrins deleted for the actin binding (ΔABD) and for the binding domain to phosphatidylethanolamine, which lies adjacent to the binding sites for the PM-scaffold protein ankyrin on the spectrin repeats 14 and 15[41,42]. Therefore, a unique deletion mutant was generated (ΔPE/ANKbs, Fig. 6a). By using FRAP, the ΔABD mutant displayed an increased mobile fraction (0.87) and a decreased half-time recovery ($t_{1/2} = 24.56$ s) as compared to the FL construct (mobile fraction = 0.74, $t_{1/2} = 41.7$ s) (Fig. 6c–e, Supplementary Table 3). The ΔPE/ANKbs mutant displayed mildly decreased half-time ($t_{1/2} = 56.8$ s), while the mobile fraction was comparable to the FL-βII-spectrin (0.73). Similar results to ΔPE/ ANKbs were obtained for the phosphatidylserine-binding mutant of spectrin (ΔPS, see Supplementary Table 3). The individual GFP-tagged PE/ANKbs domain displayed a fast diffusive behavior through the lipid bilayer as expected from an individual domain bound to a PM scaffold (Fig. 6c–e, inset). On the other hand, ΔABD and ΔPE/ANKbs mutants were all correctly targeted to the PM and excluded from actin stress fibers (Supplementary Fig. 5), as they likely incorporate into tetrameric complexes with endogenous αII-spectrin. The increased half-time recovery of ΔPE/ANKbs mutant suggests the relevant contribution of membrane interactions (PE-Ankyrin) in the diffusion modality of the spectrin-based membrane skeleton, but not in the overall spectrin meshwork mobility that is instead governed by actomyosin. Indeed, the ABD is critical for spectrin meshwork stabilization but not its localization, while potential cooperative mechanisms by different lipid- and protein-binding domains ensure PM targeting.

When looking at the cell-shape remodeling mechanisms during spreading, ΔABD-expressing cells underwent a normal P1 phase, while several collapses of protrusions were observed during the contractile phase (P2, Supplementary Movie 5 and 6). In accordance with our FRAP results, spectrin meshwork cohesion appears instrumental to sustain PM when contractility is at play, by potential binding through short actin filaments, hard to detect but also known to maintain spectrin cohesion in erythrocytes (Supplementary Movie 5 and 6). These collapsing events were different from the retractions described earlier where spectrin replaced the actin-based support to the lipid bilayer. Here, simultaneous collapses of actin and spectrin were observed (Fig. 6f and Supplementary Fig. 5E, F), followed by further attempts of the cell to spread over the substrate. As a consequence, we recorded a negative $\Delta$area time$^{-1}$ rate since positive values were offset by negative events (during 10 min of

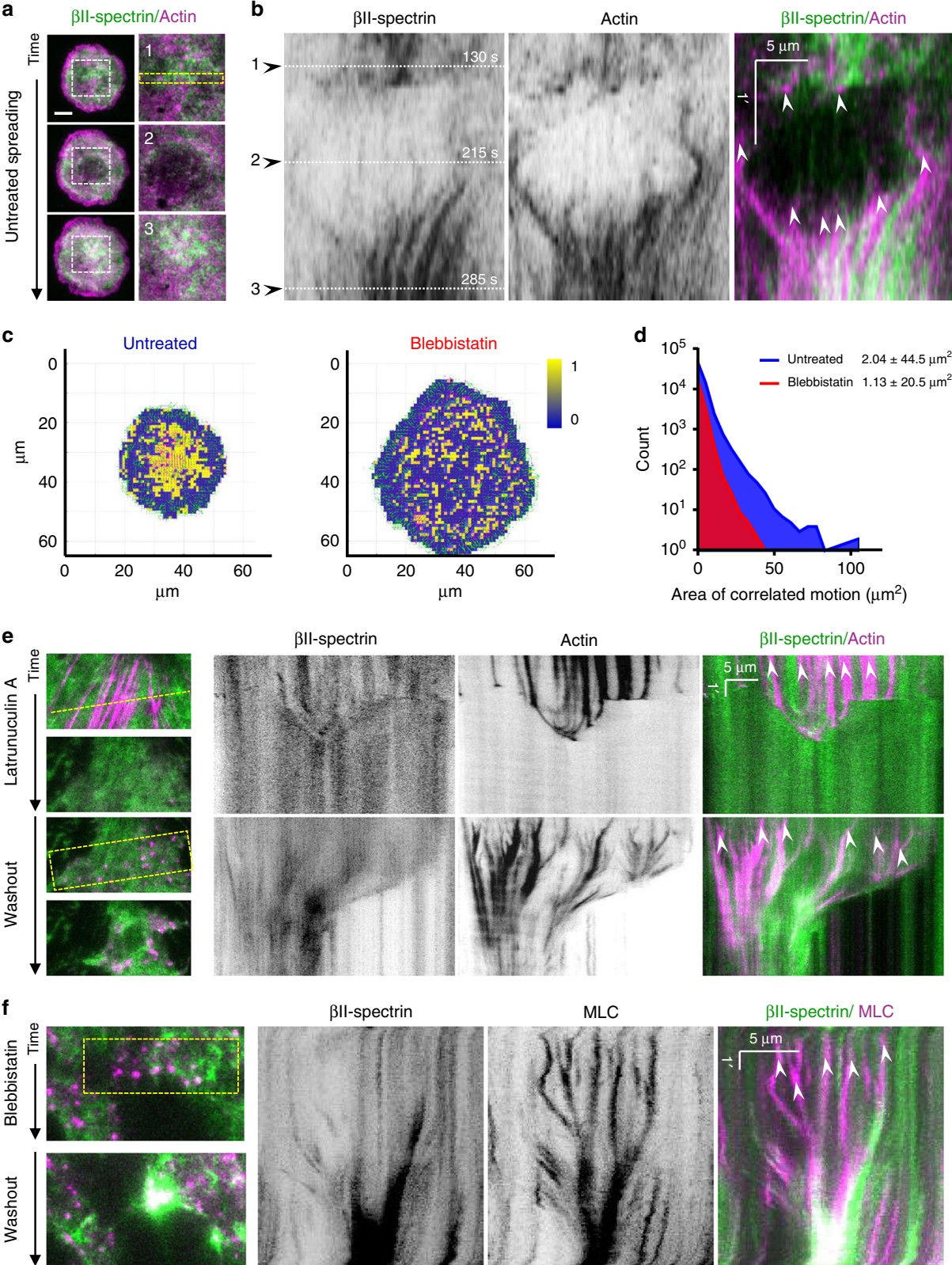

spreading after the transition in P2, Fig. 5f, g). On the contrary, FL-βII-spectrin expressing cells displayed a stereotypical steady increase in area during P2; ΔPE/ANKbs-expressing cells spread even faster than the FL-expressing cells and retraction zones highlighted a remarkable ΔPE/ANKbs-spectrin accumulation (Supplementary Fig. 5, Supplementary Movie 5), in agreement with the slower turnover rate measured by FRAP.

**Spectrin, actin, and PM interplay during mechanoresponse.** If the spectrin meshwork condensation represents a general mechanism to preserve cell and PM integrity (as shown in Supplementary Fig. 5 for cell-shape changes), it may also display similar dynamic behavior under environmentally driven mechanical perturbations. To test this hypothesis, MEFs were seeded on a deformable silicone membrane and polygonal cells,

**Fig. 4 Actin nodes drive spectrin reorganization. a** Cell spreading analysis at the cell body (zooms corresponding to the dashed white boxes), displayed by live TIRFM images (green: GFP-βII-spectrin, magenta: RFP-actin, scale bar: 10 μm). Relevant events observed between independent experiments are shown (1–3), in particular endogenous actin node formation and correspondent βII-spectrin behavior. **b** Kymographs generated in the region highlighted by dashed yellow rectangle; black arrowheads (1–3) indicate the respective images shown in (**a**) ($n = 7$ cells from independent experiments were analyzed). White arrowheads indicate the appearance of actin nodes (white arrowheads): synchronous condensation and expansion of the two non-colocalized proteins is highlighted by the coordinated side motion. **c** Two representative images of correlated PIV analyses are shown for the two experimental conditions: cells untreated and treated with 50 μM blebbistatin. In yellow are shown areas of high angular and speed correlation between βII-spectrin and actin. No correlation is shown in the blue zones (binary LUT). **d** Count distribution of "Area of correlated motion" (yellow patches) is shown in the graph: untreated cells (blue, $n = 7$ cells) displayed higher mean area and larger distribution (±SD) compared to blebbistatin-treated cells (red, $n = 5$ cells). **e** Representative images during latrunculin A and subsequent washout experiment visualized by live TIRFM (green: GFP-βII-spectrin, magenta: RFP-actin, $n = 15$ independent experiments). Kymographs are generated in correspondence of dashed yellow line and rectangle, respectively (scale and time bars are shown, white arrowheads indicate actin node formation). **f** The same experimental washout protocol is repeated with blebbistatin and representative images are shown (green: GFP-βII-spectrin, magenta: RFP-myosin light chain, $n = 4$ independent experiments). Kymographs generated in correspondence of dashed yellow box. Similar to endogenous actin node formation during spreading, coordinated motion is observed between the two channels in the absence of colocalization (timescale and scale bar are reported in the kymographs, white arrowheads indicate actin node formation).

characterized by the presence of long arcs between adhesive protrusions, were monitored during biaxial stretch on a custom-built device. Specifically, we imposed a sequential 5%-step size increase in stretch, up to 30% of the initial area; at each step, a single frame was recorded to document cell and protein behavior (Fig. 7a, b, only images at 0/10/20/30% stretch are reported, see Methods). Under stretch, GFP-βII-spectrin-expressing cells retained most of the prominent adhesions onto the substrate, while actin-treadmilling activity at lamellipodia was blocked (visualized by wide-field recording of Lifeact-RFP, Fig. 7b, ** asterisks), as we reported[43]. Consistent with our previous observations, βII-spectrin signal sharpened in retracting negative curvatures under progressive stretch, creating a frame around the cell that disappeared when the stretch was released (Fig. 7b). Further insights were also provided by the brutal detachment of weak adhesions upon the same sequential stretching protocol. Differently from the cell-driven retractions observed during spreading, actin and spectrin scaffolds condensed simultaneously and colocalized in the collapsed zones (Supplementary Fig. 6A, B). Notably, this was the only mechanical event leading to apparent colocalization between the two skeletons. We interpreted that in those particularly fast events, actin and spectrin meshworks react passively as opposed to all the other mechanical perturbations where the crosstalk between the two leads to fine-tuned adaptation and reorganization.

Next, we monitored spectrin meshwork dynamics under compressive stress. We engineered a device to apply longitudinal/uniaxial cell compression by a pneumatic PDMS piston (see Methods). In detail, six cycles of 2′ compression and 2′ relaxation at increasing pressure were performed (Fig. 7c–g, Supplementary Fig. 6C–E, Supplementary Movie 7). The first compression step defined the pressure value sufficient to engage the piston surface with the cell roof; during the following cycles, the piston was lowered at increasing pressure by the arbitrary unit. We consistently observed the nucleus to act as a dissipating additional piston on the cell cortex facing the coverslip during compression, inducing most of βII-spectrin and actin reactions to occur underneath this organelle in relatively flat MEFs (Fig. 7c, d, Supplementary Movie 7). As compressive strain increased, βII-spectrin fluorescence visualized by TIRFM increased right under the nucleus (Fig. 7c and Supplementary Fig. 6E). Simultaneously, an unexpected βII-spectrin- and actin-depleted rim formed in correspondence of the compressed nuclear envelope (Fig. 7c, Supplementary Fig. 6F). Remarkably, de novo actin polymerization characterized by concentric inward flow specifically occurred in this bare PM region within 2 min of compression (Fig. 7c and Supplementary Movie 7). Careful examination showed βII-spectrin tethered by few fibrous stretches across the rim

(Supplementary Movie 7). The release of the compressive stress stalled actin polymerization and was followed by a fast disappearance of the actin speckles (Fig. 7c, Supplementary Movie 7). Upon relaxation, βII-spectrin reacted differently than actin, since it immediately recoiled back to the precompression configuration, entangling and fencing the few remaining actin speckles (Fig. 7c and Supplementary Movie 7). Occasionally, a similar βII-spectrin clearance could also be observed by compression–relaxation of large cytosolic vesicles (Supplementary Fig. 6D). Co-staining of PM indicated that the membrane kept its integrity during the entire compressive protocol (Fig. 7f). These results represent a direct experimental demonstration of our previous observations on the dynamic cooperation between actin and spectrin in the cortex under mechanical challenges. Indeed, spectrin acts as an elastic continuous meshwork that can be stretched and depleted locally, thus working as a fence for the actin skeleton. On the other hand, bare PM is not a stable condition, and the spectrin/actin cortex is constantly trying to occupy cytoskeletal-free space by covering it like a fluctuating elastic veil-like structure (spectrin) or by polymerizing on it (actin).

The increase in intracellular pressure occasionally caused the formation of blebs (Supplementary Fig. 6F, G). In those cells, blebs induced by compression (i-bleb) clearly displayed a flow of cytosolic actin directed into the new structures, while the majority of the βII-spectrin signal was retained in the cell body. Upon the release of the piston, i-blebs were resorbed into actin-enriched tubular-like structures devoid of βII-spectrin. Here the spectrin meshwork can be uncoupled from the PM, potentially preserving cell cohesion, while actin flows into the bleb and progressively polymerizes into defined structures, as previously reported[44].

Finally, to study more directly the spectrin elastic behavior in supporting the PM, osmotic shocks were applied to the cells as a third paradigm of environmentally driven mechanical perturbation. These experiments aimed to simulate cycles of stretch relaxation of the PM. Mean fluorescence- intensity changes were simultaneously recorded for βII spectrin and a fluorescent PM marker over the projected cell area. βII-spectrin fluorescence alone, registered by TIRFM, showed reduction during hypotonic shocks and increase during isotonic relaxation; however, the ratio between βII-spectrin/PM signals did not significantly shift from the initial ratio during several subsequent cycles (Supplementary Fig. 7A, B). When soluble GFP was used as nonmembranous control, consistent reduction in the GFP/PM ratio during hypotonic shocks could be recorded (Supplementary Fig. 7C, D). Interestingly, ratiometric images failed to display homogeneous intensities throughout the entire cell, suggesting zonal enrichment or depletion of one of the two components.

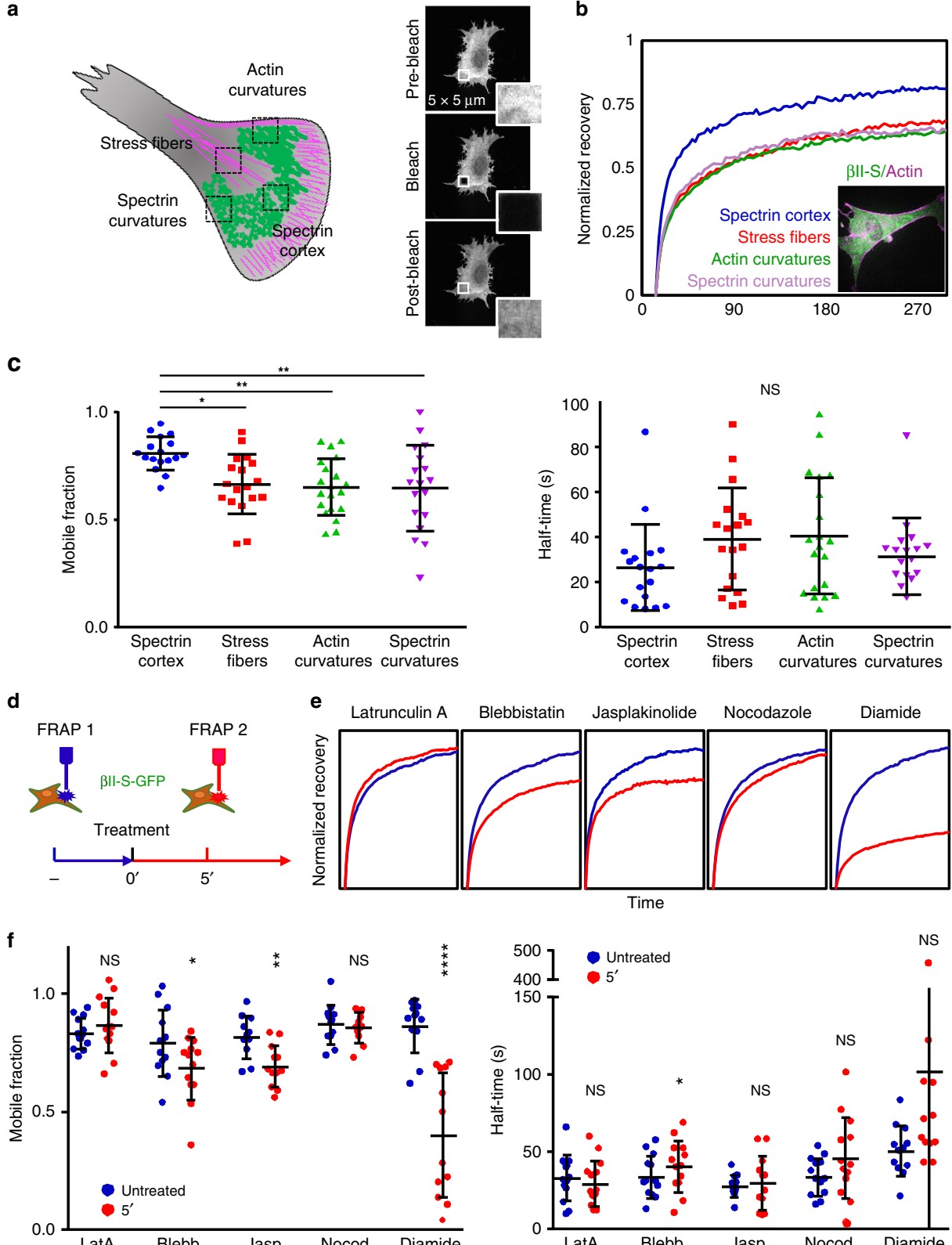

Subcellular local analysis during osmotic shocks displayed an initial reduction in the βII-spectrin/PM ratio that was compensated during later shocks in one region, while a second region of the same cell matched the linear ratio shown over the entire cell projected area (Supplementary Fig. 7E, F). Active lamellipodia during isotonic recovery behaved as expected, displaying reduced βII-spectrin/PM ratio compared to the adjacent cell body

(Supplementary Fig. 7G). Conversely, under hypotonic shock, an increase in PM tension abruptly blocked lamellipodia activity (Supplementary Fig. 7G) as expected[45,46].

Altogether, these results support the existence of local redistribution mechanisms of the spectrin-based membrane skeleton. We concluded that βII-spectrin elastic support of the PM at the whole-cell level is maintained by keeping constant the

**Fig. 5 βII-spectrin mobility displays zonal dependency and relies on myosin-II-dependent contractility. a** Schematic representation of different cell zones highlighting βII-spectrin and actin complementarity. Differential zonal βII-spectrin dynamics was analyzed by FRAP in double-transfected MEFs: RFP-Actin marked the different regions, circular (5-μm diameter) or squared (5 × 5 μm). ROIs were bleached and fluorescence recovery followed for 3 min post bleach. The same cell was simultaneously analyzed for at least 2 of the 4 different zones. The resulting normalized recovery curves are presented in (**b**), together with a representative MEF considered for the analysis (all the 4 zones under investigation were present). **c** The resulting mobile fractions and half-time recoveries are plotted in the two graphs: spectrin cortex (blue), stress fibers (red), actin curvatures (green), and spectrin curvatures (purple) (statistical analysis: one-way Anova with multiple comparisons, *$p = 0.014$, **$p = 0.0038$, $n = 18$ cells). **d** Schematic representation of the dual-FRAP assay of GFP-βII-spectrin expressing MEFs, performed before (blue) and after 5 min of treatment (red). Normalized recoveries for each drug treatment are presented in (**e**. **f**) Mobile fractions and half-time recoveries are plotted in the graphs for each drug tested: untreated (blue) and after 5′ treatment (red) (statistical analysis: paired Student' s $t$ test, two-tailed, *$p < 0.05$, **$p = 0.0021$, ****$p < 0.0001$, $n = 12$ cells). All data are presented as mean values ± SD.

ratio between the two components, while it can locally and transiently drift to allow the occurrence of specific PM-linked events.

**Endocytosis integration in the spectrin/actin/PM composite.**
Spectrin dynamics and the complementary interplay with actin pointed out the ability of the two meshworks to create PM microdomains[47]. Spectrin has been associated with PM organization, potentially positioning CME events at cell–cell junctions[9]. We tested whether spectrin was also involved in this mechanism in fibroblasts by providing molecular details of the clathrin pit distribution and dynamics in fixed and live specimens. By immunostaining analysis of endogenous clathrin heavy chain (CHC), βII-spectrin and actin, we found that the three molecular components were not colocalized and appeared rather mutually excluding each other (Fig. 8a, Supplementary Fig. 7H). To strengthen this observation, multiple discrete clathrin structures were selected, registered in 2 × 2-μm ROIs, and clustered into two groups of different size (<300 nm² and 300–500 nm²) following a recently published approach[48]. These density maps displayed clathrin pits positioned at the center of spectrin-depleted zones surrounded by spectrin-rich areas. Remarkably, the diameter of the averaged spectrin-depleted zones almost matched in size the clathrin pit projections (Fig. 8b, c). While most of the high-intensity actin structures, such as stress fibers, were clearly distinct from the pits, the averaging of >100 pits led to the identification of a discrete actin enrichment in correspondence of the clathrin staining (Fig. 8a–c). These observations are fully consistent with the current maturation models of endocytic structures[49,50]. Our analysis indicates that a potential hindrance mechanism might be at work, and the spectrin meshwork is able to delimit the zones of the assembly of clathrin pits.

Live imaging analysis confirmed the exclusion of GFP-βII-spectrin from clathrin structures visualized by the adapter mCherry-AP2σ. Specifically, AP2-decorated pits appeared in the void patches that characterized the spectrin meshwork (Fig. 8d–f). Notably, the membrane-bound PE/ANKbs domain of βII-spectrin used here as negative control did not display the same behavior. CME is a highly dynamic and heterogeneous process with several layers of regulation, including membrane tension modulation by osmolarity[51]. Therefore, we monitored this process with respect to the dynamic spectrin reorganization during osmotic changes. As expected, decreased βII-spectrin signal intensity during hypotonic shocks was accompanied by a fast-transient increase and followed subsequently by severely reduced AP2 intensity, which was restored after the transition to isotonic conditions (Fig. 8g, Supplementary Movie 8). More interestingly, kymograph analysis revealed local coordinated flow between the two channels during cell adaptation motion, suggesting that AP2 pits are fenced by the spectrin meshwork. We compared discrete AP2 pits in cellular zones characterized by high and low GFP-βII-spectrin densities during osmotic changes. Interestingly, when the βII-spectrin

reorganized into large condensation zones, several pits disappeared from the TIRF plane, most likely engulfed by the fencing capability of the spectrin meshwork (Fig. 8h, Supplementary Movie 8). This was not observed in low spectrin-density zones of the same cell, indicating that a local hindrance mechanism might be operative. We further performed in unperturbed fibroblast lifetime analysis of AP2 pits by an automated particle detection protocol from live TIRFM time lapses (see Methods): within the same cell, AP2 pits formed less frequently (AP2 counts: high density 99 ± 30, low density 144 ± 31) and displayed slower mean lifetime at zones by high βII-spectrin density compared to those at low βII-spectrin density (52.5 ± 51 s at high density vs 41.8 ± 30 s at low density, Fig. 8i, j). These results support a fencing mechanism by βII-spectrin where concentration as well as condensation of the meshwork affects local endocytic capacities. Furthermore, we investigated AP2 pit lifetime in cells that overexpressed the βII-spectrin variants previously analyzed. Overall, a mild effect was observed for all those variants compared to full-length βII-spectrin (Fig. 8k), but longer lifetime was observed in the presence of βII-spectrin ΔPE/ANKbs (62.4 ± 60 s vs 55.2 ± 51 s for FL; $n > 20{,}000$ pits in >9 independent cells). This suggests that the membrane-binding site of spectrin dominates in the regulation of endocytosis, whereas the actin-binding site of βII-spectrin is more important during motility processes linked to actin cytoskeleton.

**Discussion**
Here, we provide a universal view on how the ubiquitous and evolutionary conserved βII-spectrin dynamically interplays with actomyosin, the lipid bilayer, and the endocytic machinery to sustain the PM during intrinsic and extrinsic mechanoadaptative events (see the model in Fig. 9).

Our analysis of a variety of mammalian cells growing under various geometrical constraints, suggests the existence of discrete PM territories supported either by an actin scaffold or by a spectrin-based membrane skeleton, which likely comprises only short and hard-to-detect actin filaments. Dynamic studies in fibroblasts spreading onto adhesive substrates unveiled the assembly mechanism governing spectrin meshwork organization during the early phase of cell–substrate interaction. Upon activation of actomyosin contractility, coordinated motion of βII-spectrin and coalescent actin nodes emphasize the interplay between the two scaffolds during full maturation of the cytoskeleton (Fig. 9, cortex cycle 1–3). Despite the highly conserved actin-binding domain (ABD) that characterizes the spectrin superfamily members, our results support a model of restricted spectrin-binding capability to a subpopulation of short actin filaments adjacent to the PM. So far, such structures composed of ≈13–15 G-actin monomers have been only described in erythrocytes, and are reported to be only mildly perturbed by latrunculin A[52–54]. Consistently, we found this drug to weakly affect spectrin turnover in fibroblasts. Indeed, spectrin was excluded from actin stress fibers and protrusive actin structures,

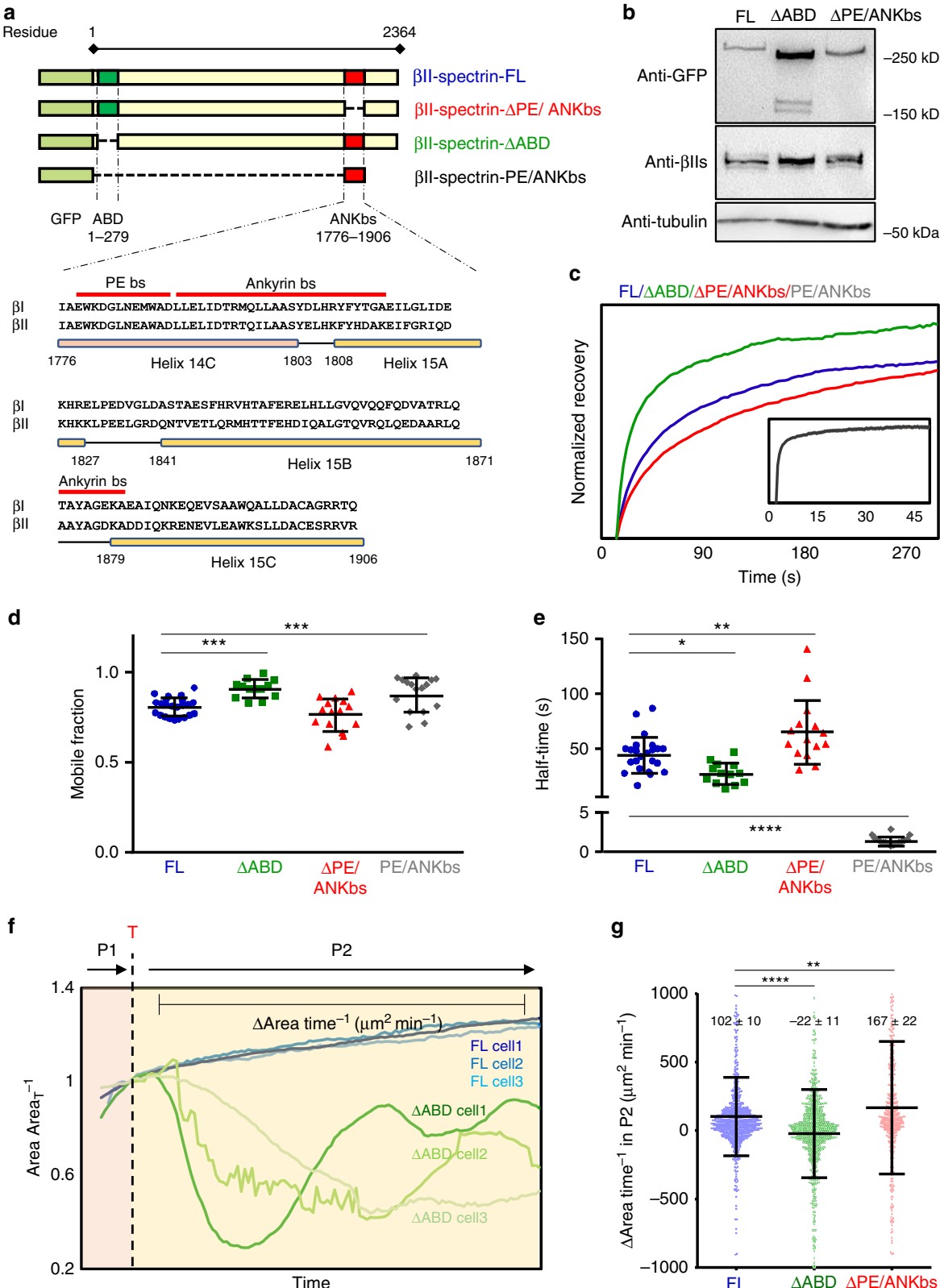

such as lamellipodia and filopodia, under all different experimental conditions tested. This level of specificity likely requires the use of accessory protein[55–57] interaction with specific phospholipid domains[58] and/or dynamic association–dissociation mechanisms of spectrin–actin bonds instrumental for membrane stability and trafficking[59].

We observed spectrin meshwork dynamics to rely on myosin-II contractility during cytoskeletal maturation as well as in established cell cortex. Current models describing the transmission of myosin-dependent contractility at the isotropic cortex hardly explain how forces exerted on nonpolarized scaffolds can produce homogeneous movement. Since short actin filaments might be too small and rigid to generate coherent contractility[60],

**Fig. 6 βII-spectrin variants show different dynamic properties. a** Schematic representation of the βII-spectrin deletion variants analyzed in this study. In light green is shown the GFP tag, in dark green the actin-binding domain (ABD), and in red the PE/Ankyrin-binding site on spectrin repeats 14 and 15. Sequence alignment of PE/ANKbs between βI and βII-spectrin is shown: the mapped binding sites (red lines) are highly conserved between the two proteins. **b** Total cell lysates of MEFs expressing exogenous GFP-βII-spectrin variants analyzed by anti-GFP and anti-βII-spectrin antibodies in western blot assay ($n = 2$). **c** FRAP analysis of βII-spectrin deletion variants expressed in MEFs. In the graph are reported the normalized recovery curves (PE/ANKbs in the inset), while mobile fractions and half-time recoveries are plotted in the two graphs (**d**–**e**) (statistical analysis: one-way Anova with multiple comparison, $*p = 0.0163$, $**p = 0.0071$, $***p < 0.005$, $****p < 0.0001$, data are presented as mean values ± SD, $n = 22$ (FL), 13 (ΔABD), 15 (ΔPE/ANKbs), and 16 (PE/ANKbs) cells. **f** Normalized cell area growth during P2: three stereotypical MEFs transfected with GFP-βII-spectrin FL are plotted in blue, while MEFs expressing ΔABD are shown in green and followed for 10 min after P1/P2 transition (T) by live TIRFM. **g** Quantification of ΔArea time$^{-1}$ extracted from subsequent frames in time lapses during 10 min after transition into P2 (FL $n = 792$ frames, ΔABD $n = 840$ frames, ΔPE/ANKbs $n = 502$ frames, $n = 7$–9 independent cells. Data are presented as mean values ± SD, statistical analysis one-way Anova with multiple comparison, $**p = 0.0031$, $****p < 0.0001$).

the hierarchical actin/spectrin organization and the cohesiveness provided by the described meshwork have the potential to reconcile this paradox. This is highlighted by the expression of the spectrin ΔABD mutant, documenting edge instability characterized by synchronous spectrin/actin retractions (Fig. 9, zone III). We speculate that in the actin-poor but spectrin-rich lamella, the spectrin meshwork can act as a force-transmitting veil-like structure underneath the PM. This veil creates a potential mechanical continuum at the lamellipodia/lamella border with the contractile structure localized deeper in the cell body. The dominant-negative ΔABD expression might, thus, uncouple these distinct frameworks and create a mechanical discontinuity in this cohesive architecture. However, also long-term perturbation of the other cytoskeletal systems, such as microtubules, affects the organization and the cohesiveness of the spectrin meshwork and associated proteins[9], albeit with different mechanisms and time-scale that require further investigations.

In cells with established architecture and fully engaged with the substrate, spectrin is organized into a fibrous assembly covering PM territories with different morphological properties. Our results during extrinsic mechanical perturbations suggest that spectrin works as a sum larger than its individual parts (dimers and tetramers) and reacts differently, depending on the nature of the applied mechanical cues. We provide further support to a mesoscale-fencing mechanism brought about by macromolecular network condensation upon mechanical stimuli, first proposed in neuronal axons under compressive and bending forces, rather than a molecular stretch/relaxation model based on intramolecular distance discrepancies between different EM and super-resolution studies[25,61].

Altogether, our results highlighted the opposite polarity between the spectrin-based membrane skeleton and actin. To the best of our knowledge, this is the first dynamic and mechanistic description of actin/spectrin dualism during cell-shape remodeling events. Based on different mechanical perturbations of the PM, we propose that spectrin is maintained globally at a constant ratio with the lipid bilayer. Locally, it responds instead to intrinsic cues, such as PM collapse (Fig. 9, zone VII), or to external perturbations, such as cell–cell fusion[27]. These responses point toward a meshwork condensation rather than de novo recruitment, as a self- and cell-protective mechanism. The actin/spectrin-coordinated dynamics is particularly exploited during spectrin cortex clearance induced by mechanical compression of the cell, which triggered actin nucleation in response to spectrin displacement (Fig. 9, nuclear compression cycle); this observation frames the event that anticipates the distinct spectrin and actomyosin accumulation during shear deformation described in ref. [27]. Together with spectrin exclusion from protrusive lamellipodia (Fig. 9, zone I), these results suggest the existence of a potential interference mechanism that hinders actin polymerization in the presence of a spectrin-enriched cortex.

Our observations during different environmental perturbations strongly support the existence of a non-Brownian diffusion mode of the spectrin skeleton along the PM[62]. Indeed, the spectrin meshwork defines PM microdomains able to constantly remodel in response to external and internal cues. Such capacity integrates well into the revised three-tiered mesoscale version of the fluid–mosaic model of PM organization[47,63], and can complement the so-called picket-and-fencing mechanism prominently led by the actin cortex. Here, we implemented this model by adding the spectrin-rich territories in the context of membrane dynamics and topological organization. As previously hypothesized from biochemical data[9], we observed membrane-trafficking events, such as CME, taking place at PM microdomains hamstrung by βII-spectrin, while pit maturation sustained by actin polymerization occurred specifically within βII-spectrin fenestration (Fig. 9, cortex cycle 1–3). Of note, several mechanisms have been identified in the regulation of CME, many of them showing discrepancies and controversy with one another[64]. Such a role in positioning PM- trafficking events is consistent with a recent report proposing spectrin as a general ruler for cell–cell adhesion molecule organization in neurons[20]. Moreover, CME lifetime seemed influenced not only by the different βII-spectrin densities, but also by the meshwork capability to interact with the PM through PE and/or ankyrin bindings. We propose that the highly dynamic composite nature of the cortex under mechanoresponses is mainly regulated by an orchestrated menàge a 4 between PM, spectrin, endocytosis, and actomyosin contraction–polymerization (see our conclusion model Fig. 9). More generally, these results indicate that the spectrin-based membrane skeleton dynamics is critical to shape and coordinate many PM-linked cellular processes in physiology and pathology.

## Methods

**Cell culture and media**. Immortalized MEFs derived from RPTP $\alpha + /+$ murine background (Su, Muranjan, and Sap, 1999) were grown in complete media composed of DMEM (Lonza) supplemented with 10% Fetal Bovine Serum (FBS South America, Euroclone) and 2mM L-glutamine at 37 °C and 5% $CO_2$. Cell density never exceeded 70% confluency to favor single-cell instead of tissue-like behavior. For imaging experiments, MEFs were seeded on borosilicate glass coverslips of 1½ thickness (Corning) or Nunc Glass Base Dishes (Thermo Fischer Scientific) coated with sterile 10 μg ml$^{-1}$ fibronectin (Roche). During live imaging experiments, the media was exchanged 30 min before experiments in serum-deprived $Ca^{2+}$-buffered Ringer's solution (see Supplementary Table 4 for media composition) to avoid $Ca^{2+}/Mg^{2+}$ withdrawal shocks and phenol-red background contamination during light excitation. For experiments monitoring clathrin dynamics, Ringer's solution was supplemented with 10% FBS. Other cell lines were obtained from IFOM Cell Bank and cultured as follows: NIH3T3, HS27, RPMI, and A-431 were cultured in DMEM (Lonza) supplemented with 10% FBS. For MCF-10A, DMEM-Ham's F12 (Biowest VWR) was supplemented with 5% Horse Serum (Life Technologies), 10ug/ml insulin, 20 ng/ml EGF, 500 ng/ml hydrocortisone, 100 ng/ml cholera toxin, and 2mM L-glutamine; MDA-MB 231 cells were cultured in Leibovitz's L15 (Biowest VWR) supplemented with 10% FBS and 2mM L-glutamine; HeLa cells were cultured in DMEM, 10% FBS, 1 mM sodium pyruvate, 0.1 mM Non-Essential Amino Acids solution, and 2mM L-glutamine. HUVEC primary cells were cultured

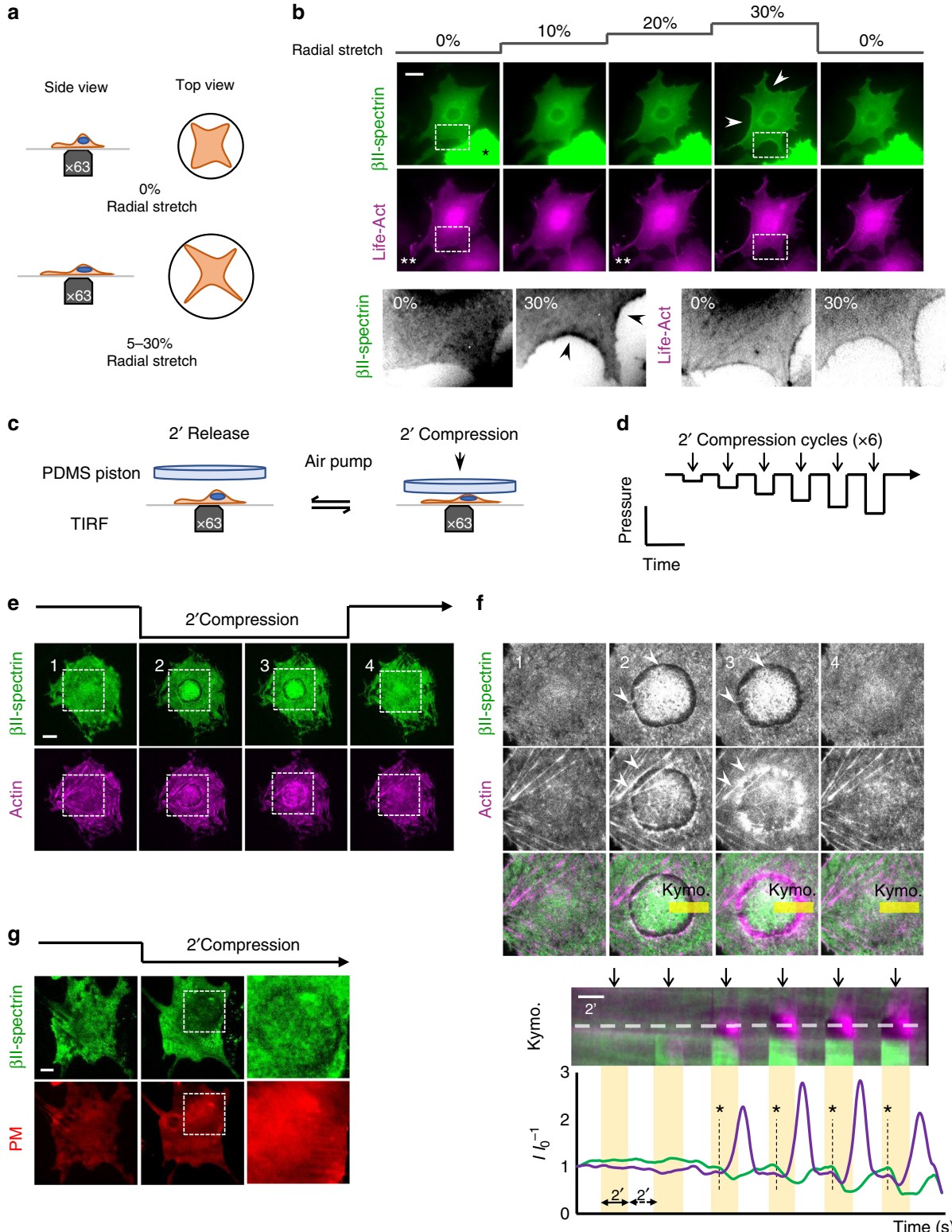

in all-in-one ready-to-use Endothelial Cell Growth Medium (Cell application Inc. Merck). Reagents' supplier and identifier are listed in Supplementary Table 4.

**Plasmids and transient expression**. Plasmids for mammalian transient expression used in this study are listed in Supplementary Table 4, describing the original source and identifier. To generate GFP-βII-Spectrin full length, the human gene *SPTBN1* (gene ID: NM_003128) was amplified by pcr and cloned into the pEGFP-C3 backbone (Clonetech) between HindIII and SacII restriction sites, with the fluorescent tag located at the N terminus interspaced by an additional flexible

linker composed of 11 residues (KYSDLELKLAA). For the generation of GFP-βII-Spectrin-ΔPE/ANKbs, residues 1776–1906 were deleted from full-length βII-Spectrin. The same peptide was cloned in-frame with GFP into the pEGFP-N1 backbone to generate GFP-PE/ANKbs only, between HindIII and SacII restriction sites. For the generation of GFP-βII-Spectrin-ΔPS, residues 421–530 were deleted from full-length βII- Spectrin. For the generation of GFP-βII-Spectrin-ΔABD, residues 280–2364 were amplified from full-length GFP-βII-Spectrin, and cloned into pEGFP-C3 backbone between HindIII and SacII restriction sites. Other plasmids were purchased or obtained from external sources listed in Supplementary Table 4. Transient expression was obtained by electroporation performed 24 h

**Fig. 7 βII-spectrin reactions to mechanical perturbations highlight the interplay with actin. a** Cartoon representation of the cell-stretching device implemented in this study. **b** MEFs transfected with GFP-βII-spectrin (green) and LifeAct-RFP (magenta) seeded on the fibronectin-coated silicone membrane and stretched biaxially (sequential 5% step-size increase, up to 30%, representative images of $n = 16$ cells in 3 independent experiments). After each step, the cell under investigation was recentered and refocused; representative images by EPI-fluorescence microscopy are shown (white asterisk indicates transfected MEFs with high intensities excluded from the analysis, scale bar: 20 µm). Double white asterisks highlight a lamellipodia blocked during the stretching. In the dashed boxes 1–2, representative cell-edge behavior observed among independent experiments, highlighting peculiar condensation of βII-spectrin at curvatures not enriched by actin (arrowheads) at maximal stretch (30%). $n = 9$ cells in 2 independent experiments. **c** Cell compression setup and the applied step-increase protocol (**d**) are schematized. **e** GFP-βII-spectrin (green) and RFP-actin (magenta) expressing MEFs are imaged by live TIRFM during the entire compressive protocol. Four relevant time points are shown: precompression, early and late compression, and during the release phase (scale bar: 10 µm, $n = 13$ cells in 4 independent experiments). Key details (precompression 1, early and late compression 2–3, and release 4) consistently observed between independent experiments are highlighted by dashed boxes and zoomed in (**f**). Reaction in correspondence of the nuclear edge, brought into the TIRF plane by the compressive stress (white arrowheads), is quantified by the kymograph analysis (**c**, bottom) over the yellow rectangle. Fluorescence intensities across the dashed line in (**f**) are plotted in the graph; clearance of βII-spectrin and the delayed actin polymerization is observed (asterisks and dashed lines in the graph). **g** Control cells expressing PM marker (red) and βII-spectrin (green) were subjected to the same compression/relaxation protocol. Insets focused on the cortex underneath the nucleus, where PM marker retains its continuity (scale bar: 10 µm, $n = 2$ independent experiments).

before the experiment using the Neon electroporation system (Thermo Fisher Scientific). For each transfection, $1 \times 10^6$ cells were trypsinized, washed once with PBS, and mixed with a total of 10 µg of recombinant DNA in electroporation buffer R (Invitrogen). Cells were singularly electroshocked at 1600 mV for 20 ms by placing the electroporation tip into the column filled by E2 buffer (following the manufacturer's specifications). After the shock, cells were immediately seeded onto tissue-culture-grade plastic dishes filled with complete culturing media and allowed to recover for at least 24 h.

**Confocal and TIRF microscopy.** Confocal microscopy was performed on a Leica TCS SP8 laser-scanning confocal module mounted on a Leica DMi 8 inverted microscope, equipped with a motorized stage, and controlled by the software Leica Application Suite X (ver. 3.5.2.18963). For image acquisition, a HC PL APO CS2 63 ×/1.40 oil immersion objective was used. DIC, epifluorescence (EPI), and total internal reflection fluorescence microscopy (TIRFM) of fixed specimens, live time lapse of spreading cells, drug treatments, osmotic shocks, cell stretching (EPI mode only), and cell compression were performed on a Leica AM TIRF MC system. Two different TIRFM-grade objectives were used: HCX PL APO 63 ×/1.47NA oil immersion and HCX PL APO 100 ×/1.47NA oil immersion. Three different laser lines were used for fluorochrome excitation: 488 nm, 561 nm, and 635 nm. A specific dichroic and emission filter set for each wavelength have been used. The microscope was controlled by Leica Application Suite AF software (ver. 2.6.1.7314), and images were acquired with an Andor iXon DU-8285_VP camera. For live imaging experiments, environmental conditions were maintained by an Okolab temperature and $CO_2$ control system.

**Micropatterning by quartz mask.** Borosilicate glass coverslips (Corning) were washed with pure isopropanol, airy-dried, and activated by a plasma cleaner (Harrick Plasma) for a few minutes. The surface was passivated by incubation with 0.1 mg ml$^{-1}$ poly(ethylene glycol)-b-poly(l-lysine) (PEG–PLL, Ruixibio) for 1 h at room temperature to prevent surface coating by adherence factors such as fibronectin. A quartz mask (Delta mask B.V.) was washed with 70% ethanol and activated under UV light using a UV lamp (UVO Cleaner, Jelight). PEGylated coverslips were aligned to the desired pattern in the mask, which then was illuminated under UV light for 7 min. This step is critical since the mask prevents UV illumination of the passivated surface, while burning the PEG–PLL in correspondence of the desired pattern. After UV light exposure, those patterns were specifically coated with 10–30 µg/ml fibronectin for 1 h at room temperature, while the passivated surface prevented fibronectin adherence. After rinsing the coverslips several times with PBS, $1–5 \times 10^4$ MEFs were seeded and cultured at 37 °C in the same media described before.

**Micropatterning by PDMS stamping.** Polydimethylsiloxane (PDMS) was prepared by mixing the two components of the Sylgard 184 silicone elastomer kit (Dow Corning) in 1:10 ratio and degassed. It was then poured on the mold with the desired patterns printed by lithography (MBI Singapore), degassed again, and cured overnight at 65 °C. PDMS was carefully peeled off the mold and plasma cleaned to make the surface hydrophilic. The stamps were then coated with 10–30 µg ml$^{-1}$ fibronectin for 30 min at room temperature. Excess of fibronectin was air-dried; the stamps were gently pressed onto the previously silanized borosilicate glass base dishes or coverslips for 1 min and then carefully removed. Silanization was achieved by incubating coverslips in a methanol solution containing 0.16% v/v silane (Sigma-Aldrich) for 1 h, followed by three washes in pure methanol. To passivate the rest of the surface and prevent cell attachment over the nonstamped area, coverslips or glass base dishes were treated with a solution of 0.1 mg ml$^{-1}$

PEG–PLL (Ruixibio) for 1 h at room temperature. After the incubation, the dishes were rinsed several times with PBS and $1–5 \times 10^4$ MEFs were seeded and cultured at 37 °C in the same media described before.

**Immunofluorescence.** The antibodies used in this study were the following: mouse anti-βII-spectrin (dilution 1:200, BD-bioscience), rabbit anti-βII-spectrin (1:200, Abcam), mouse anti-αII-spectrin (1:200, Invitrogen), rabbit anti-β-actin (1:100, Cell Signaling), and mouse anti-clathrin heavy chain (1:500, clone X22, Thermo Fisher Scientific). Before fixation, cells were seeded on 10 µg ml$^{-1}$ fibronectin-coated coverslips/glass base dishes. Cells were fixed in 4% paraformaldehyde for 10 min, then neutralized using 10 mM $NH_4Cl$ in PBS for 10 min. Alternatively, fixation was performed in ice-cold pure methanol for 2 min at −20 °C. Cells were subsequently washed three times with PBS (5 min each), permeabilized for 2–5 min using PBS containing 0.1% Triton X-100, and blocked with 3–5% BSA for 10 min at room temperature. Cells were incubated with primary antibody for 1 h at room temperature or overnight at 4 °C. After 3 washing steps in PBS, cells were incubated with Alexa 488/647-conjugated goat anti-mouse or anti-rabbit (1:100–1:400, Thermo Fischer Scientific) and Alexa 488/568-conjugated phalloidin (1:200, Sigma-Aldrich) for 1 h at room temperature. After three washes in PBS of 5 min each, cells were mounted with anti-fade glycerol-based media (for confocal microscopy) or PBS (for TIRF microscopy) and stored at 4 °C. All primary antibodies and fluorophore-conjugated secondary antibodies are listed in Supplementary Table 4. For intensity correlation analysis (Pearson's coefficient and scatterplot), the Fiji plug-in JACoP was used[65] and plotted using R ggplot2 package.

**Western blotting.** For western blot analysis, cells were lysed directly on the plate by adding opportune amount of modified Laemmli sample buffer composed of Tris-HCl 135 mM (pH 6.8), sodium dodecyl sulfate (SDS) 5%, urea 3.5 M, NP-40 2.3%, β-mercaptoethanol 4.5%, glycerol 4%, and traces of bromophenol blue. Total protein content was normalized by seeding cells at equal densities; this lysis buffer does not allow total protein quantification, but prevents membrane-bound proteins from being degraded during trypsinization. Equal volume between samples was then loaded onto 12–8% SDS polyacrylamide gels and transferred after electrophoretic separation onto nitrocellulose membrane (Amersham GE-Healthcare). After the transfer, membranes were blocked in PBS supplemented with 0.1–0.3% Tween20 and 5% milk for 1 h at room temperature, then incubated overnight at 4° with primary antibodies at the following dilutions: mouse anti-βII-spectrin 1:2000 (BD-bioscience), rabbit anti-βII-spectrin 1:2000 (Abcam), and mouse anti-β-tubulin 1:5000 (Sigma-Aldrich). After three washing steps in PBS–Tween20 (0.1–0.3%), membranes were incubated for 1 h at room temperature with HRP-conjugated secondary antibodies (BioRad). Three washing steps of 5 min in PBS–Tween20 (0.1–0.3%) were performed between primary and secondary antibody incubation. Proteins were detected by ECL Western blotting reagents (Amersham GE-Healthcare), using the digital Chemidoc XRS + system run by the software Image Lab (Biorad).

**Fixed spreading assay.** MEFs were seeded on 10 µg ml$^{-1}$ fibronectin-coated glass base dishes in complete media, as previously described. A total of $1 \times 10^5$ cells were seeded and fixed at different time points (between 5 and 20 min after seeding) by 4% paraformaldehyde diluted in PBS for 10 min. Fixed cells were incubated with the fixable membrane dye FM4-64 FX (Thermo Fischer Scientific) according to the manufacturer' specification (2.5–5 µg ml$^{-1}$). A second fixation step was applied to preserve cell membrane staining and avoid diffusion of the dye during immunostaining protocol. Cell permeabilization was performed in PBS–Triton X-100

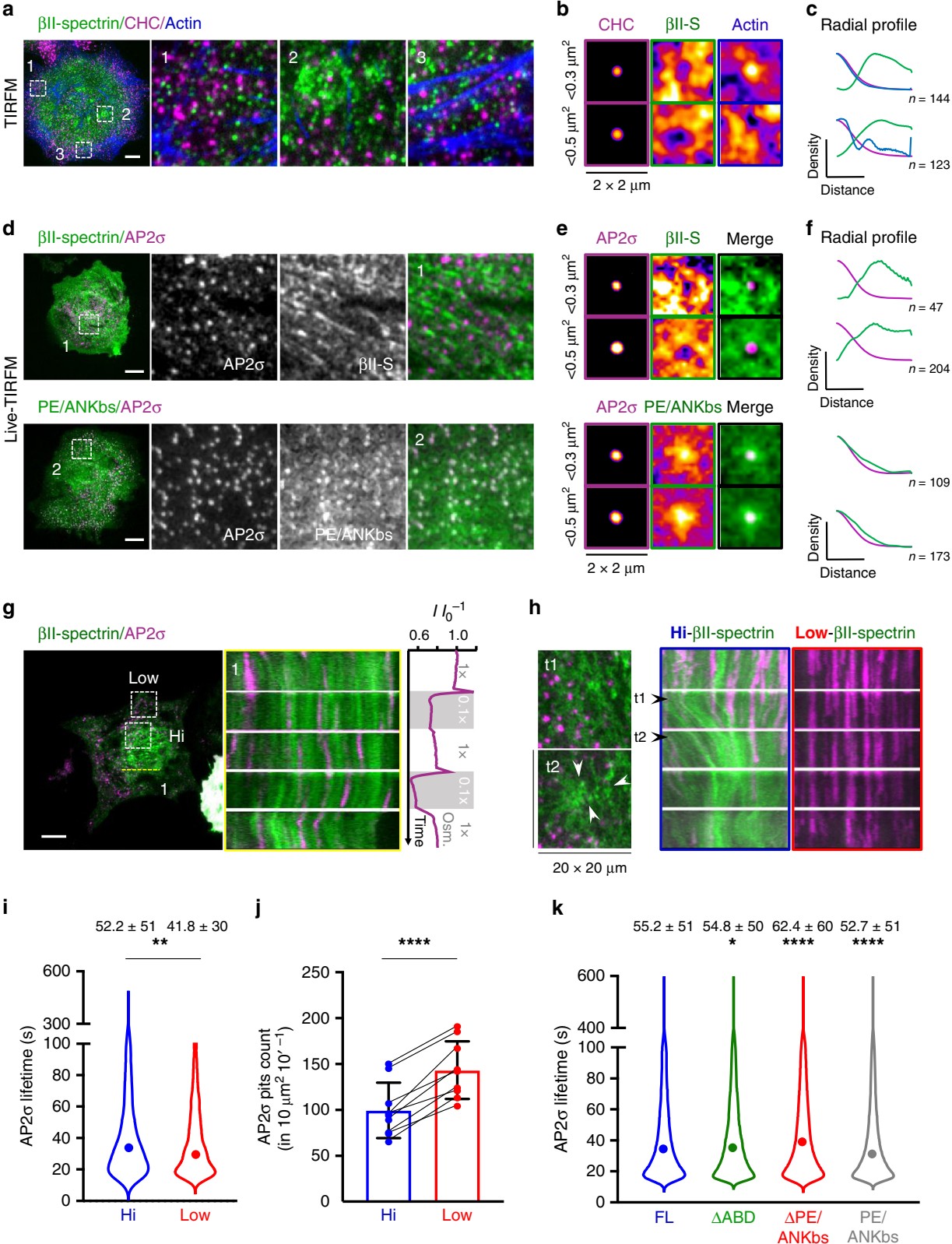

(0.1%) for 1–2 min. Immunostaining for βII-spectrin and F actin (phalloidin) was performed as previously described for TIRFM investigation. From the raw images, the background signal was subtracted, and edge-preserving filters were applied to the FM4-64 fluorescence signal to generate a binary mask of the projected cell area. The Fiji built-in "Analyze particle" tool was then used to extract the projected area as well as the integrated densities of βII-spectrin and F-actin. Linear regression analysis was then performed by the software R, and raw data plotted by the software GraphPad Prism.

**Live spreading assay**. The spreading assay was performed on custom-designed 2-way aluminum slides, sealed on the two planar faces by 22 × 22-mm glass coverslips welded by high vacuum grease (Sigma-Aldrich). Coverslips were previously acid-washed with a 20% $HNO_3$ solution for 2 h at room temperature, followed by a final wash in pure acetone before being dried and coated with 10 μg ml$^{-1}$ fibronectin for 1 h at 37 °C. The chamber was rinsed with 1× Ringer's solution and equilibrated on the microscope stage at 37 °C. The top and bottom glass surfaces of the chamber allow simultaneous fluorescence and DIC illumination during media

**Fig. 8 Integration of clathrin endocytosis in the spectrin/actin/plasma membrane composite. a** TIRF microscopy images of MEFs immunostained for endogenous βII-spectrin (green), clathrin heavy chain (CHC, magenta), and F actin (blue) (scale bar: 10 μm, images are representative of many cells, immunolabeled in $n = 2$ independent experiments). Three different subcellular regions are highlighted (dashed boxes 1–3): high-density CHC zone (1), βII-spectrin-rich zone (2), and actin stress fibers (3). **b** Density maps generated by aligning discrete clathrin pits of small (<0.3 μm$^2$, $n = 144$) and larger size (0.3–0.5 μm$^2$, $n = 123$) are shown, while in (**c**) correspondent radial intensity profiles for the three channels are presented. **d** Representative live TIRF microscopy images of MEFs transfected with GFP-βII-spectrin (green) and the clathrin adapter AP2σ-mCherry (magenta) (scale bar: 10 μm, $n = 9$ and $n = 8$ cells), unsharp and Gaussian filters were applied. PE/ANKbs-transfected fibroblasts show homogeneous PM localization and enrichment in correspondence of AP2σ pits, as shown by density maps (**e**) and radial profile analysis (**f**) generated as previously described (GFP-βII-spectrin: <0.3 μm$^2$, $n = 47$ and 0.3–0.5 μm$^2$, $n = 204$; PE/ANKbs: <0.3 μm$^2$, $n = 109$ and 0.3–0.5 μm$^2$, $n = 173$). **g** Representative images of TIRFM time lapse during osmotic shocks (scale bar: 20 μm, $n = 6$ independent cells): kymograph is generated in correspondence of the dashed yellow line (1). Whole-cell AP2 intensity signal in response to the osmotic shocks is plotted in the graph (magenta, vertical plot). **h** In total, 20 × 20-μm ROIs drawn at low- and high-spectrin-density zones around AP2 pits (correspondent to dashed boxes in (**g**), only zooms of Hi-density zone are reported at two different frames). Kymographs display the differential behavior of the pits observed during spectrin remodeling (representative of $n = 6$ independent cells). **i** Violin plots (dots = median values) of AP2 pit lifetime at high and low βII-spectrin-density zones of 10 μm$^2$ (statistical analysis paired Student's $t$ test, one-tailed, $p = 0.0094$, mean values ± SD are reported in numbers, $n = 505$ (Hi), 756 (Low) pits in 9 independent cells). Total AP2 pit counts during 10 min of recording over the same zones are reported in (**j**); measurements within the same cell are highlighted by the connecting line (statistical analysis paired Student's $t$ test, two-tailed, $p < 0.0001$, data are presented as mean values ± SD, $n = 9$ cells). **k** Violin plot (dots = median values) of AP2 lifetime in MEFs expressing different βII-spectrin variants (statistical analysis one-way Anova with multiple comparison, *$p = 0.0112$, ****$p < 0.0001$, mean values ± SD are reported in numbers, $n = 23,000–29,000$ pits in 8–10 independent cells).

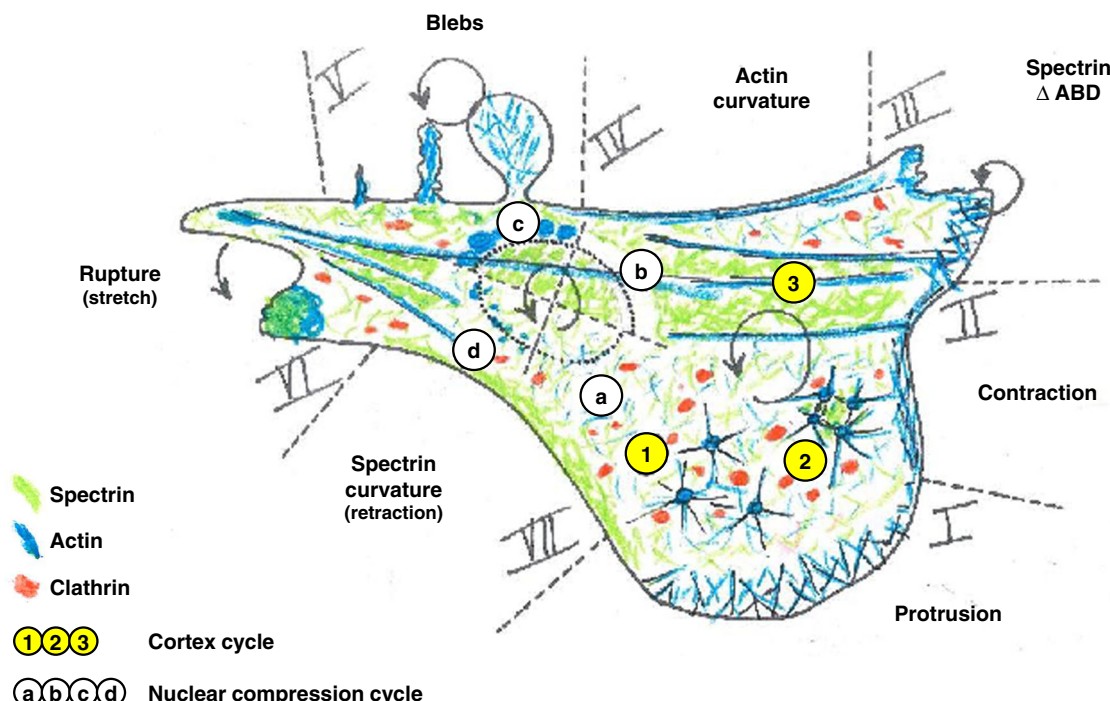

**Fig. 9 Model resuming our findings on the dynamic response of βII-spectrin during mechanoresponse.** I–VII highlight the protein behavior at cell periphery. I Upon protrusion, actin polymerization dominates. II During contraction, actin is condensed and forms transverse arcs. In I and II, βII-spectrin is secluded and "passively" follows actomyosin lead. III Deletion of the actin-binding domain of βII-spectrin induces edge instability upon contraction activation. IV Mature actin bundles sustain the PM; βII-spectrin is not recruited to those actin curvatures. V Upon cell compression, blebs enriched with actin but devoid of βII- spectrin are formed. While actin polymerizes and condenses in the bleb, βII-spectrin localizes and marks the precompression position of the cell edge. VI Abrupt cell detachment induces a "plug-like" process in which actin and βII-spectrin seem to colocalize. VII In actin-depleted but spectrin-rich edge curvatures, βII-spectrin condenses as the edge moves inward, potentially holding the PM and responding to the increased load. a–d cycle highlights the cell cortex behavior under nuclear compressive stress. In a–b, the native acto-spectrin cortex get cleared under the edges of the nuclear envelope upon compression. c During compression, in this belt of cleared membrane, actin polymerization occurs and fills up progressively the bare bilayer. d Upon relaxation, spectrin meshwork elastically recoils, entangling the polymerizing actin and restoring the original cortex organization. (1)–(3) cycle highlights the cell cortex behavior. (1) In spectrin-less zones, actin nodes can form. (2) These aster-like structures move and coalesce upon myosin-II-mediated contractility. This mechanism modifies spectrin meshwork local density synchronously, with expansion and condensation between coalescing nodes. Further actomyosin condensation leads to bundle formation interleaved by spectrin-rich territories. (3) These high-density zones impede clathrin-mediated endocytosis (red dots), which otherwise occurs between spectrin fences. Those steps occur upon cell control or in response to drugs/mechanical stimuli.

addition/exchange. MEFs were transfected 24 h before the experiment with the opportune plasmid combinations. Before the experiment, cells were trypsinized, centrifuged for 5 min at $300 \times g$, washed once with PBS, and serum-starved in suspension for 30 min at 37 °C in $CO_2$-independent 1× Ringer's solution. Suspended cells were thereafter kept at room temperature up to 3 h. For each time lapse, $1$–$5 \times 10^4$ cells were fluxed into the imaging chamber to optimally observe single-cell spreading; for this reason, cell aggregates or debris were carefully avoided during imaging. The time lapse started after a positively double-transfected cell engaged with the fibronectin-coated coverslip; cells were then followed for 15–20 min at a frame rate of 2–5 s frame$^{-1}$. For experiments in the presence of myosin-II inhibitor blebbistatin (Sigma-Aldrich), cells were suspended in 1× Ringer's solution supplemented with blebbistatin at 50 µM final concentration; imaging chambers were filled with the same 1× Ringer's solution supplemented with blebbistatin to avoid rebound effects when cells were fluxed into the imaging chamber. The correct cell behavior was monitored by DIC acquisition, in particular by focusing on cell integrity, isotropic spreading in P1, lamellipodia formation, and buckling. The fluorescent channels were analyzed as described in the next few sections for cell edge and cell body behavior.

**Spectrin and actin intensities at the cell edge**. For cell-edge analysis, a custom macro in Fiji was written. The background signal was subtracted, while edge-preserving filters were applied to the actin fluorescence signal to generate a dynamic binary mask of the cell area over time. The centroid of the cell was used as a reference point to identify each angle between 0° and 359° on the cell outer circumference. From the edge, the signal was eroded by 25 pixels (≈3.2 µm at the resolution of ≈130 nm pixel$^{-1}$), and mean intensity of the fluorescent signals within this 25-pixel wide belt was computed into the final kymographs composed of 360 pixels on the x axis, one for each angle, while the y axis represents the total number of frames. Speed of the cell edge was extrapolated from the distance variation between the outer edge and the cell centroid, at known pixel size and time frame. Values were considered positive when the edge moved away from the centroid and negative when it moved closer. For comparisons between independent cells, actin- and spectrin-intensity measurements were normalized to 1 at null local speed. The results were analyzed and plotted with the ggplot2 R package.

**Correlated spectrin and actin flow velocity analysis (PIV)**. Correlation between speed and directionality of the two fluorescent channels was performed on the same live dataset analyzed for cell-edge dynamics during spreading. The background signal was subtracted. Particle Image Velocimetry (PIV) was performed independently on single fluorescent channels by a custom macro in Fiji, excluding from the analysis the portion of the cell close to the edge (−50 pixels) and the frames corresponding to the initial spreading phase P1 (≈50 frames). The resulting vector fields of the two channels (i.e., RFP-actin and GFP-βII-spectrin) mapped both the speed magnitude and directionality (angle) of the channels independently. To identify synchronous angular and speed correlation between the two channels, binary values were assigned as follows: value of 1 was assigned to areas where the speeds of the two channels were within ±50% of each other, and where the directions shared the same quadrant (i.e., when the difference of angle was less than $\pi\ 2^{-1}$). Areas that did not meet these criteria were assigned a value of 0. The fraction of correlated flow was then given as the ratio between the area covered by correlated velocities and the total area of the cell. The area of correlated motion was calculated for each binarized frame by the "Analyze particle" tool, and by randomly applying the Watershed algorithm to segment neighbor areas with no morphological continuity. The same parameters were blindly applied to untreated and blebbistatin-treated time lapses to avoid bias. The results were analyzed and plotted with the ggplot2 R package.

**Drug administration and osmotic shocks**. MEFs were transfected 24 h before imaging with the opportune pair of constructs as previously described. Cells were trypsinized, seeded on glass coverslips coated with 10 µg/ml fibronectin ($10^3$ cells) for 2–3 h in complete media before imaging. The media was replaced with $CO_2$-independent 1× Ringer's solution and mounted on a 2-way imaging chamber that allows on-stage media exchange as previously described for the spreading assay. Time lapse consisted of an initial phase of 5 min where sufficient frames were acquired at steady state (internal control); the media was then replaced by 1× Ringer's solution supplemented with the opportune drug. Addition of media exceeded the volume of the imaging chamber to avoid drug dilution and rebound effect. Specifically, 1 µM latrunculin A, 10 µM blebbistatin, 1 µM jasplakinolide, 5 µM nocodazole, and 5 mM diamide were singularly used (Sigma-Aldrich). Perturbed cells were then imaged for 30 min at 1–5 min frame$^{-1}$. Intensity calculations were carried out in Fiji by subtracting the background signal and creating a dynamic binary mask of the GFP-βII-Spectrin signal; the Fiji built-in "Analyze particle" tool was applied to obtain mean fluorescence-intensity values at different time points. Only untreated, 5- and 30-min treatment time points were plotted using the software GraphPad Prism. For experiments that required osmotic shocks, MEFs were treated following the same procedure described above, but time lapses were obtained at higher temporal resolution (2–5 s frame$^{-1}$). At the given time points (every 3–5 min depending on the experiment), the media was replaced with hypotonic 0.5× or 0.1× Ringer's solution, exceeding the volume of the imaging

chamber to avoid incomplete media exchange. Mean intensity calculation was done in Fiji by subtracting the background and creating a dynamic binary mask to monitor changes in fluorescence intensity over time. In the case of ratio measurements between the two fluorescent channels, mean intensities of the first two frames of the meaningful channels were averaged and arithmetically matched to obtain the initial ratio value of 1 (nonstoichiometric). Indeed, the purpose of these experiments was to calculate the fluctuation in content more than a stoichiometric measurement between the two fluorescent proteins. Intensity data were then averaged, analyzed, and plotted using the software GraphPad Prism.

**Cell stretching**. Cell stretching experiments were performed using an automated cell-stretching dish (International patent: WO 2018/149795). The components of the cell-stretching dish were designed using SolidWorks software and 3D printed using a stereolithography-based 3D printer (Form 2, Formlabs) coupled with an autoclavable and biocompatible dental resin (Dental SG resin, Formlabs). The printed parts were rinsed in isopropyl alcohol for 5 min to remove any uncured resin from their surface, and then post cured in a UV box to finalize the polymerization process and stabilize the mechanical properties. The printed parts were then polished and assembled to create the lower (cell chamber) and the upper portion (aperture driver) characterizing the stretching dish. Before the experiments, the components of the lower portion were autoclaved and assembled to clamp a deformable silicone membrane (thickness 0.005″, SMI) to four anchoring clips, one for each quadrant, thus creating a cell culture chamber (Supplementary Fig. 7A). The aperture driver mobilizes simultaneously these four quadrant clips to perform biaxial stress/strain. Specifically, the strain field has been tested to be uniform in the center region of the dish (within 6 mm$^2$) by labeling the membrane with a marker during preliminary validation; to make sure that the strain was consistent between independent experiments, only cells in the center of the membrane (within 6 mm$^2$) were imaged. The silicone base of the dish was coated with fibronectin (10 µg ml$^{-1}$) at 37 °C for 1 h. A total of $10^5$ MEFs were seeded in the cell chamber and allowed to spread in complete media for 3 h at 37 °C. Before imaging, the stretching dish was assembled by connecting the upper portion (driving unit) to the cell culture chamber, and complete media was exchanged with 1× Ringer's solution. Biaxial stretching was applied directly on the stage of the Leica AM TIRF MC microscope; fluorescence images were acquired in the EPI-fluorescence mode since silicone impedes the physical principles of TIRF microscopy. The imaging protocol was designed as follows: a 5% step-size increase in the biaxial stretch was applied; fluorescence images were acquired in both fluorescent channels (excitation wavelengths: 488 and 546 nm) after each step, up to 30%, which is the maximum stretch allowed by the system, followed by a final image acquisition after relaxation to 0%. The duration of each step was dictated by the manual refocusing and recentering of the cell under investigation, and the time required for image acquisition. Due to these technical limitations, continuous imaging was not possible, and on average, each step lasted for ≈60 s.

**Cell compression**. Cell compression experiments were performed using a cell compression device (International patent: WO 2019/086702) capable of applying dynamic compression stress to single cells. The main components of the compression device were designed using SolidWorks software and 3D printed using a stereolithography-based 3D printer (Form 2, Formlabs), following the same procedure described for the cell-stretching device. The cell compression device consists of an air chamber connected to an air-pressure regulator. Before the experiments, a polydimethylsiloxane (PDMS, Sylgard 184) piston was microfabricated to have circular pillars (200 µm in diameter) on its surface, and attached to a deformable silicone membrane (thickness 0.005″, SMI) through plasma bonding. The membrane with the piston was then clamped to the air chamber. The assembled cell compression device was connected to the air-pressure regulator and then locked to the cell culture dish (seeded with cells) through mechanical ribs (Supplementary Fig. 7B). In particular, a total of $10^5$ cells were seeded specifically on 27-mm Ø Nunc Glass Base Dishes (Thermo Fisher Scientific), exchanged to 1× Ringer's solution 30 min before imaging, and the original Petri lid replaced by the compression device. A dynamic compressive load was applied to cells by increasing the air pressure inside the compression device through the pressure regulator, thus controlling the movement of the membrane and the piston to compress the cells underneath (Supplementary Fig. 8B). DIC and TIRF illumination was used to monitor cell reaction at 5 seconds/frame rate. ROIs to measure fluorescence-intensity fluctuation were drawn in Fiji, while the DIC was used to monitor the engagement of the piston with the cell roof. As the compression strain increased, the cell became flatter, and DIC imaging decreased its contrast in flat specimens. Due to cell height variation, the device does not allow precise absolute readout of the pressure applied to cells, being cell deformation a noncontrollable variable. For this reason, two different approaches were applied. The first one was intended to cause maximal cell response: compression pressure was slowly increased until bleb formation was observed. For i-bleb/cortex ratio, mean fluorescence intensities in the two channels were obtained in the projected bleb area, and divided by the mean fluorescence intensities in the adjacent cortical region of similar area. The second approach was designed to better control the applied strain: the initial compression was thus set at the first value required to engage the piston with the top of the cell as monitored by DIC. This first step hardly caused a reaction detectable by TIRF at basal cell level, but allowed consistency between independent experiments; for this

reason, six different cycles of compression (2 min) and relaxation (2 min) were performed at increasing pressure by arbitrary units, as schematized in Fig. 6d, e; the duration of the compression step (2 min) was chosen to allow adaptive mechanisms to occur and avoid long-term detrimental effects on cell integrity. Intensity calculations were carried out in Fiji by subtracting the background signal and drawing ROIs across the perinuclear rim formed during compression. Prestretch mean intensity for each single compressive events was divided by maximal mean intensity registered during the subsequent 2-min compressive step. Data were plotted using the software GraphPad Prism.

**Frap experiments**. MEFs expressing GFP-βII-spectrin constructs were imaged 24 h after transfection with a confocal spinning disk microscope (Olympus) equipped with iXon 897 Ultra camera (ANDOR) and a FRAP module furnished with a 405-nm laser. The environmental control was maintained with an OKOlab incubator. Images were acquired using a 100×/1.35Sil silicone oil immersion objective. MEFs were trypsinized and seeded on glass base dishes (Matek, Sigma-Aldrich) coated with 10 μg ml$^{-1}$ fibronectin. Before imaging, $CO_2$-independent 1× Ringer's solution was exchanged. Squared regions of interest of 5 × 5-μm length were photobleached with the 405-nm laser at 100% intensity, and post-bleach images were acquired with 15–20% laser intensity for 100 frames (1 frame every 3 s for full-length and truncated GFP-βII-spectrin constructs and every 0.5 s for PE/ANKbs only). FRAP data were analyzed, and curves fitted to the one-exponential recovery equations (one-phase association) by the software GraphPad Prism:

$$I = I_0 + I\max \times \left[1 - e^{-(k)(t)}\right], \quad (1)$$

where $I$ represents the relative intensity compared to the prebleach value, k the association rate, and t the half-time recovery expressed in seconds.

For the FRAP assay on different cortical zones, cells were double-transfected with GFP-βII-spectrin and RFP-Actin. The fluorescent actin channel was used to discriminate between different cytoskeletal structures: spectrin cortex, stress fibers, and actin- and spectrin-rich curvatures.

For the dual-FRAP assay, cells were seeded on a fibronectin-coated glass coverslip and mounted on the 2-way imaging chamber that allows on-stage media exchange as described in the section "Live spreading assay" and osmotic shocks. The first FRAP measurements were conducted in serum-free 1× Ringer's solution; at completion, the media was exchanged and independently supplemented with the cytoskeletal drugs at the concentrations previously implemented. Specifically, 1 μM latrunculin A, 10 μM blebbistatin, 1 μM jasplakinoldie, 5 μM nocodazole, and 5 mM diamide (Sigma-Aldrich). Cells were allowed to equilibrate with the new media for 5 min; afterward, a second FRAP analysis was performed on the same cell and the same ROI previously analyzed.

**Clathrin pit-density maps**. TIRFM images were acquired as previously described. For analysis of fixed specimens, 2 × 2-μm ROIs were selected by segmenting discrete pits of random sizes, not overlapping with neighboring structures (clathrin plaques by heterogeneous shape were not considered for the analysis). All images were registered and stacked in FIJI for pit-size calculation. CHC/AP2 images were upscaled by a factor of 10, Yen autothreshold applied to create a binary mask, and particle size calculated by the "Analyze particle" tool in FIJI. ROIs were divided at this point according to the size that took into account the diffraction limit of TIRFM (<300 nm$^2$ and 300–500 nm$^2$); particle size greater than 500 nm$^2$ was not considered informative. Raw images of the two clusters were then independently stacked and z-projected for median-intensity values. Gaussian blur filter (1-pixel radius) and a scale factor of 10 were applied to the projected images to homogenize the signals. Projected images of the two particle-size groups were then matched for signal intensities between the corresponding channels. Normalized radial plots were generated by the Radial Profile plug-in (FIJI) on the final 2 × 2-μm projected images.

For live imaging datasets, Unsharp Mask (1-pixel radius, 0.6 weight) and Gaussian blur filter (1-pixel radius) were applied to the raw images before ROI selection. No additional filters were applied to the final projected images. Similar filtering strategy was applied in live datasets during osmotic shocks.

**Pit lifetime analysis**. MEFs were transfected as previously described with the opportune combination of plasmids: GFP-βII-spectrin variants and mCherry-AP2σ. Cells were seeded on glass base dishes coated with 10 μg ml$^{-1}$ of fibronectin in complete media, and allowed to spread for 2 h. Before imaging, media was replaced by 1× Ringer's solution supplemented with 10% FBS, to avoid serum deprivation. TIRFM time lapses were acquired for both fluorescent channels for 10 min at frame rate of 4 s frame$^{-1}$. Laser power and camera settings were kept constant between independent experiments (20–30% and 300-milliseconds exposure), and only cells with comparable fluorescence intensities were recorded. For automated AP2 pit tracking, images were processed blindly by the custom-written macro: background signal was subtracted (50.0 pixel radius), LUT set to grayscale at 8-bit rate. The "Mosaic" tool on Fiji was used to detect particles by applying constant settings: Kernel radius 3 pixels, cutoff 0.001, percentile signal/noise 0.5, and link range 1 frame. Detected particle lifetime was calculated by the software R; only particles that were continuously detected for at least 16 s (4 subsequent frames) were plotted by the software GraphPad Prism. No subsequent

additional filtering on the particle lifetimes was performed. For high/low spectrin-density calculation, 10 μm$^2$ ROIs were drawn arbitrarily at the region of maximum and minimum intensities in the GFP-βII-spectrin channel within the same cell, carefully avoiding cell periphery.

**RNA extraction and qPCR analysis**. Cells were cultured as previously described. At least 2 $10^5$ cells were lysed, and RNA was extracted with the RNAeasy Mini Kit (Qiagen) following the manufacturer' specifications. About 1 μg of RNA was ret-rotranscribed using "qScript cDNA Synthesis kit" (Quantabio). For gene expression analysis, 5 ng of cDNA was amplified (in triplicate) in a reaction volume of 10 μL containing the following reagents: 5 μl of TaqMan® Fast Advanced Master Mix, 0.5 μl of TaqMan Gene expression assay 20× (Thermo Fisher). The entire process (retrotranscription, gene expression, and data analysis) was performed by the qPCR-Service at Cogentech-Milano, following the ABI assay ID Database (Thermo Fisher). Therefore, only gene ID of the spectrin murine genes analyzed could be provided here (Mm01315345_m1, Mm01180701_m1, Mm01326617_m1, Mm00661691_m1, Mm01284057_m1, and Mm01239117_m1). Murine *SPTBN*5 primers were custom- designed (Fw: GGACGCCAGTGTTCACCAA Rev: GCCCCCTTGTAGCAGCTT) since they were not implemented in the database. Real-time PCR was carried out on the 7500 Real-Time PCR System (Thermo Fisher), using a pre-PCR step of 20 s at 95 °C, followed by 40 cycles of 1 s at 95 °C and 20 s at 60 °C. Samples were amplified with primers and probes for each target, and for all the targets, one NTC sample was run.

Raw data (Ct) were analyzed with "Biogazelle qbase plus" software, and the fold change was expressed as CNRQ (calibrated normalized relative quantity) with standard error. Graphpad was used as references to normalize the data. Three independent experiments were then averaged and plotted using the software Graphdap Prism.

**Statistical analysis**. No statistical methods were used to predetermine sample size. All the graphs and plots are presented as mean ± SD, except the FRAP recovery curves that are presented as mean only. Statistical analysis by unpaired Student's *t* test was performed when the comparison between two experimental groups was required (i.e., normalized i-bleb/cortex ratio, fret before/after drug treatment). One-way Anova analysis was performed when multiple experimental groups were present. One-way Anova analysis with multiple comparisons between groups was also performed in parallel, as shown for the quantification of maximal intensity at the cell body during the sequential compression protocol. All the experiments were performed in triplicate or more, as indicated in the figure legends.

**Reporting summary**. Further information on research design is available in the Nature Research Reporting Summary linked to this article.

## Data availability
All full-scan immunoblots and all datasets used to generate each of the graphs presented in the work are provided in the Source Data file. The authors declare that all the original microscopy data that support the findings of this study are available from the corresponding authors upon reasonable request. Source data are provided with this paper.

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

## Acknowledgements

We are grateful to the IFOM imaging facility personnel, in particular D. Parazzoli, M. Garre, and E. Martini for technical support. IFOM cell culture facility and qPCR facility personnel. We thank Jagadish Sankaran (NUS) and Toh Kee Chua (NUS) for the preliminary experiments of this project. We thank MBI protein core facility personnel, in particular Chen Hongying for providing the constructs. We thank Paulina Nastaly (IFOM) for the help with micropatterning. We thank Mike Sheetz (MBI), Marco Foiani (IFOM), Sarah Barger (NYU), and all the members of Gauthier's, Scita's, and Maiuri's groups for helpful discussion. We thank Sara Sigismund (IEO) for kindly providing additional constructs. This work was supported by an IFOM starting package, a Mechanobiology Institute of Singapore grant WBS R-714-016-007-271, and an Italian Association for Cancer Research (AIRC) Investigator Grant (IG) 20716 to NCG, by H2020-MSCA individual fellowship to AG (796547) and by Fondazione Umberto Veronesi (FUV) doctoral fellowship for CG.

## Author contributions

A.G. and N.C.G. contributed to the conception and design of the experiments, interpretation of data, drafting, and critical revision of the paper. A.G., C.G., N.C.G., Q.L., F.A., and P.M. performed the experiments. Q.L. and F.A. conceived the engineered devices and helped with the experiments. A.G., N.C.G., M.A.F., G.S., and P.M. contributed to data analysis and interpretation. All authors critically revised and approved the last version of the paper.

## Competing interests

The authors declare no competing interests.
