## [Peer Review File · Nature Communications]

Reviewers' comments:

Reviewer #1 (Remarks to the Author):

This manuscript presents an interesting story of the spatio-temporal dynamics of the spectrin-based membrane skeleton during cell spreading, in relation to that of the better understood actin cytoskeleton. Among the interesting findings are that 1) dynamic assembly of the spectrin-rich membrane domains are mutually exclusive with the F-actin rich domains during cell spreading, thereby coordinating the creation of membrane territories, 2) spectrin-rich domains are associated with regions of negative membrane curvature, 3) spectrin dynamics at the membrane depends on myosin contractility, but not on actin polymerization, and 4) that spectrin-rich domains surround clathrin-coated pits and flow laterally in coordination with the endocytic pits, suggesting the spectrin domain forms a restricting mesh around the forming pits. The response of spectrin-rich domains to whole cell stretching and compression is also discussed. Overall, this study presents novel findings in an area that has not been studied in context of actin dynamics during cell spreading but would be substantially strengthened by experiments to directly test the function of the spectrin-rich membrane domains in cell spreading dynamics and/or in clathrin-mediated endocytosis. In addition, the experiments using cell stretching and compressive stress to investigate how mechanical perturbations affect the spectrin-rich domains are very confusing and not well-related to the higher resolution studies of the relationship of spectrin and actin domain dynamics in the rest of the paper. There are also some interpretations which have been oversimplified and need more clarification and exploration.

1. Figure 5 describes the dynamics of various spectrin variants (truncations, deletions, specific domains) but does not explore any functional consequences. For example, what is the effect of overexpression of the beta II-spectrin variants missing the actin binding domain or the PE binding domain on rates and morphodynamics of cell spreading and F-actin assembly dynamics? What about the effects on the localization, rates and extents of clathrin-mediated endocytosis? In the discussion, it is proposed that the spectrin creates an interference mechanism to hinder actin polymerization (line 443); this could be tested in the cells overexpressing the spectrin variants missing the ABD. In addition, it is surprising that a spectrin variant missing the ankyrin binding domain, a critical domain for membrane assembly, was not tested. This should be done. The effects of beta-II spectrin knockdown on actin polymerization dynamics during cell spreading and would also be desirable. In the

2. Figure 6 presents the effects of whole cell stretching and compression on spectrin assembly at the cell membrane of the ventral surface using a clever apparatus. However, the images presented are mostly of whole cells, so that the relationship of the so-called mesoscale domains of spectrin and F-actin are not apparent; it is very hard to relate this data to that of the previous figures. What does the image time course look like in different cell regions at high magnification? I also wonder what the compression experiment really means when the ventral membrane is being squashed up against the nucleus. This seems like an induced artifact. It would be more informative to test the role of mechanical perturbations at the mesoscale on the different domains shown in Figure 1, for example by AFM or other microscale methods. More direct information on biomechanical properties is needed before the authors can claim that the "spectrin, actin and plasma membrane create a continuous but dynamic composite material" (line 313). Alternatively, Figure 6 could potentially be left out, and these questions may perhaps be left to a future study, so that the other areas can be better addressed.

3. The authors claim to be studying membrane domains on the order of 100 μm^2 in size, but the FRAP experiments (and the mechanical ones in Figure 5) seem to be on the whole cell or much larger regions. The FRAP experiments should be performed at the appropriate scale (or if they are, this should be more clearly indicated and presented in the figures). With a 100 μm^2 domain, does it matter where the FRAP is performed in the cells with respect to their spreading morphology or at different times during spreading? The FRAP half-time data in Figure 4D are presented on a cell by cell basis, which reveals the wide variations from cell to cell after the drug treatments. However, the data should also be averaged and statistical analysis performed to evaluate significance of the differences.

It is difficult to make a firm conclusion without this. If the changes are not significant perhaps the authors need to rethink their interpretations, and/or test the FRAP in smaller sized membrane domains, as mentioned above.

4. The spectrin-rich domains presumably have actin filaments associated with them; otherwise the spectrin would not form a long-range meshwork (network). The authors refer to these as actin protofilaments; however, assuming this spectrin network is similar to other spectrin networks, they are real actin filaments, just short ones, and should be called "short actin filaments" or "short F-actin", not protofilaments which implies they are a different structure than normal actin filaments, which is not the case. However, it is worth mentioning that there is no direct evidence regarding the actual actin filament lengths in spectrin networks in any cells besides red blood cells where they are 40 nm long; the authors could make this clear as well.

Most importantly, the authors should verify that there is F-actin in the spectrin-rich domains by collecting images for a longer time and/or increasing the image intensity (ie, overexposing). The other F-actin-rich domains with stress fibers or lamellipodia will need to be blown out to see the lower abundance/density of F-actin in the spectrin-rich domains. This reminds me of the situation when studying F-actin in the cell nucleus; perhaps those approaches could provide some helpful tips here. Assuming the authors can obtain the sensitivity to visualize the F-actin in the spectrin-rich domains, it would be very interesting to perform FRAP and compare this F-actin dynamics to the spectrin dynamics and to F-actin dynamics in the other types of domains. Their observation in Figure 4 that LatA treatment did not affect the spectrin dynamics assayed by FRAP is consistent with the idea that the short actin filaments in the spectrin network are stable; as shown previously for RBC actin filaments in the red cell spectrin-based membrane skeleton by Gokhin et al., *MBoC* 26:1699, 2015.

5. The association of the spectrin-rich domains with the regions of negative membrane curvature shown in Figure 6 is interesting but should be better documented and quantified in the context of the spreading experiments- ie, what about negative curvature regions in spreading cells? Crawling cells? etc. In addition, this should be discussed in the context of previous studies of actin and myosin association with negative or positive membrane curvatures. For example, Elliott, Fischer et al. *Nat Cell Biol.* 17:137, 2015. Lou, Zhao et al. *PNAS* 116 (46):23143, 2019, among others.

6. The spectrin-rich domains should be referred to as the spectrin-based membrane skeleton, or spectrin-rich domains, not the spectrin cytoskeleton; the cytoskeleton extends into the cell whereas the key feature of the spectrin-based membrane skeleton is that it is associated with the membrane and not extending into the cytoplasm. The authors also use "membrane-associated spectrin meshwork" on line 207-this is fine too.

7. The authors conclude from several of their experiments that the spectrin meshwork dynamics confers resilience to the cell; e.g., see line 268. However, the resilience of the spectrin meshwork is not measured directly, for example, using AFM or micropipette pulling or in poking experiments at the mesoscale; see also comment #2 above.

8. The term 'mesoscale' is not clearly defined at the beginning of the manuscript.

9. line 37. Spectrin is not ubiquitous. It is only in metazoan cells.

10. lines 64-65. The original electron microscopy studies from Jeannine Ursitti should be mentioned in discussion of the lengths for the spectrin tetramers in situ in the erythrocyte membrane skeleton Ursitti et al., 1991 *Cell Motil Cytoskeleton* 19(4):227; Ursitti and Wade, 1993 *Cell Motil Cytoskeleton* 25(1):30. It might also be helpful for the authors to read the reviews on the RBC spectrin membrane skeleton to improve the context of the introduction and discussion, especially in terms of the role of spectrin in mechanoprotection (lines 70-71); e.g. from Mohandas and Evans, 1994, *Ann Rev Biophys Biomol Struct* 23:787, and Mohandas and Gallagher, 2008 *Blood* 112:3939.

11. The Lardennois reference is not correct. The Frick, Schmidt and Nichols reference is missing. There may be other errors in the reference list.

12. The antibody binding sites (epitopes) of the anti-spectrin antibodies should be stated in the results and in the methods, and their locations taken into account in the results and discussions. Some antibodies recognize the middle of the spectrin tetramer and others the ends near the actin binding site. It is conceivable that localization results may differ depending on the location of the antibody binding. Also, the locations of the GFP tags should be stated.

Reviewer #2 (Remarks to the Author):

I am reviewing the manuscript entitled 'Mesoscale Dynamics of Spectrin and Acto-Myosin shape Membrane Territories during Mechanoresponse' submitted to Nat Comm by Ghisleni, Gauthier and co-authors. The manuscript provides an in-depth characterization of the interplay between actin and spectrin cytoskeletons and uncovers several novel and unexpected features of the cortical cytoskeleton and thus complements existing, earlier work from Bennet et al. The authors show a seemingly contradictory and antiphasic role of actin and spectrin cytoskeleton during the cell spreading process but also during the response of the cell to external mechanical stimuli. They also investigate the spectrin dynamics as being dependent on myosin contractility, which requires the actin binding domain. At the same time, spectrin occupies territories distinct from actin, a fact that seemed to evade my intuitive understanding. The authors also investigate the dynamics of spectrin cytoskeleton under compression and show a striking exclusion of spectrin from the perinuclear actin belt and thus might contribute to our understanding of the cellular mechanoresponse during confined cell migration. The present topic is timely, the data is well analysed and described extremely well, and the manuscript is written clearly with display items of superior quality. I thus think the article is suitable for publication after minor revisions.

General comment:

Spectrin and actin have distinct domain during expansion and retraction and during mechanical compression. This is particularly surprising by the fact that deletion of the actin binding domain changes the properties of the spectrin network. How does spectrin do this and what is the role of the actin binding domain, if it does not colocalize with actin? Perhaps I missed that point in the manuscript but the understanding might benefit from an explicit paragraph in the discussion. In Figure 6, no details on the cell stretching experiment have been given, e.g. How long was the stretch applied for and how was the strain field quantified? I assume that cell at the center of the membrane experience more stretch than cell on the periphery? Do the cells experience the stress delivered biaxial or radial? Is the device attached to a previous publication? If not, more details are mandatory before publication. Have the authors tried to knockdown the endogenous β -spectrin in the cells? The recovery seems to be very fast in comparison to other literature values, so I might conclude the the labeled spectrin is predominantly excluded from the native network. Can the authors comment on this? Also, apart of the PH domain, β -spectrin has an ankyrin binding domain - do the authors expect a role for this?

1) Line 37: the key component spectrin; spectrin is not a single component and neither is it a single protein.

2) Line 67: Brown et al, PLOS Comp Biol, 2015 could be cited here

- 3) Line 73: instead of mechanical tension, spectrin seems to protect axons from mechanical deformation during buckling by keeping them under tension
- 4) Line 212: diamide is not a common drug. Can the authors quickly introduce what it does and why it is specific for spectrin?
- 5) Line 215: The result suggests that microtubules antagonize spectrin assembly at the plasma-membrane. Can the authors discuss this?
- 6) Figure 7 could be split and panel I deserves its own figure.
- 7) several typos throughout methods including the following ...
- 8) Line 972, monitorable = detectable
- 9) Line 1037: Graphpad reads Graphdap
- 10) Figure S3 panel C, Axis label reads Integrated instead of Integrated
- 11) The actin signal in the bleb is due to polymerized actin or diffusion of RFP-actin into the bleb?

Reviewer #3 (Remarks to the Author):

Summary:

In the manuscript entitled "Mesoscale dynamics of Spectrin and Acto-myosin shape membrane territories during mechanoresponse", Ghisleni et al. find that spectrin and acto-myosin cytoskeletons are spatially distinct but cooperate during cell mechanoresponses specifically cell adhesion and compression. To this reviewer's knowledge, this work is the first detailed demonstration of the dynamic interplay between the acto-myosin and spectrin cytoskeletons at the cell cortex. This reviewer found the writing, experimental creativity/design, and high-quality data garnered from the techniques used in this manuscript to be compelling. Further, while much attention has been focused on cytoskeletal interconnectedness between the actin and microtubule cytoskeletons exactly how these dynamics connects to the complicated interactions with lipids, vesicles, and the cell cortex has been vastly understudied. This work clearly advances the cytoskeletal crosstalk and mechanobiology fields - congratulations to the authors on this outstanding work.

Below are specific minor concerns and points of clarification:

- The complimentary pattern between Spectrin and actin in multiple human and mouse cell lines is really interesting. Mention/speculation on what molecules/mechanisms may be directly mediating this coordination in the discussion could make this work more accessible to a broader readership. Further, given the specific regions of complimentary patterning (leading edge, stress fibers, negative membrane curvature) I am curious if septins or BAR proteins mediate/contribute to such interactions.
- Many groups have started using spectrin "seeds" to assess aspects of actin dynamics in microfluidic systems. Could the complimentary pattern be mimicked in an in vitro system? Would actin polymerization be in distinct non-spectrin regions similar to the observations in cells? These experiments would likely be very challenging and require reconstituted lipids, making this suggestion well beyond the scope of this work.
- The drug treatments employed throughout this work are extremely helpful in dissecting the role of spectrin-acto-myosin crosstalk, but they are also kind of like hitting the cell with a hammer. If the data is available, siRNA knockdown of a myosin phenocopying the blebbistatin data or additional text in the manuscript about the limitations of this broad-spectrum drug could be helpful. In this similar vein, are specific components of actin assembly i.e. Arp2/3 complex (also critical for endocytosis) is regulating the actin behaviors described here. Have the authors considered using CK666/Arpin to assess the actin assembly factors possibly involved with this crosstalk?
- While there are hundreds of endocytosis events happening at any given time during image

acquisition for experiments described in Figure 7, quantification of BII-S “holes” and CHC/actin/AP2a puncta could give the reader a better idea of the frequency of these associations.

Minor formatting considerations:

- Line 486 a space is needed between plasma and membrane.
- Line 455 The word “figure” in the Figure 7I callout should be capitalized for consistency.
- Line 462, while this reviewer appreciated the “ménage a 4” callout due to similar actin-based mechanisms, perhaps a more specific way for describing the Spectrin convergence point between acto-myosin contraction, polymerization, and exo/endocytosis is more appropriate.

Reviewer #4 (Remarks to the Author):

Ghisleni et al. examined the mesoscale dynamics of cortical actin and spectrin cytoskeleton during cell cortex remodeling. The strengths of the paper include the discovery of mutually-exclusive cortical zones of actin and spectrin networks, a range of biophysical assays to show the mechano-responsive behavior of actin and spectrin networks, and quantitative image analysis throughout the manuscript. Taken together, the authors concluded that the antagonistic interplay between cortical actin and spectrin networks is required for organizing the lipid bilayer into spatially confined cortical territories during cell mechanoresponse.

Major concerns:

1. Although the observations of the complementary localization of actin and spectrin are convincing and interesting, the manuscript is largely descriptive without providing significant mechanistic insight. For example, what triggers the mutually exclusive localization of actin and spectrin? Does spectrin exhibit actin polymerization- and/or actomyosin-induced mechanosensory response? How contractility may regulate spectrin recruitment? Also, how are the AP2 pits hooked to the spectrin network?
2. It is difficult to follow the biophysical experiments (Line 269-359, Fig 6, Fig S5) for several reasons: 1) poor description of the experiments and lack of interpretation of the results, 2) inconsistent order of experiments between the main text and the figures, and 3) mislabeling of the in-text citations, figures and figure legends.
3. More experiments are required to strengthen some conclusions. For example, the investigation of the effect exerted by the actin cytoskeleton on the spectrin network is based on a single pharmacological perturbation experiment. This should be further strengthened with additional actin depolymerizing and stabilizing reagents to disrupt the actin network for more thorough analyses.
4. Despite extensive quantification in the manuscript, statistical analyses are often lacking.

Specific comment:

1. The title: The article is mainly about the mutually exclusive actin and spectrin zones (and endocytosis), hence the title needs to be revised to reflect the main findings of the work.
2. Line 34: Show clear definition of “mesoscale” in the introduction. (100nm~10um)
3. Line 56: “triangular-like” should be “triangle-like”
4. For Fig. 1A inboxes on the right, Fig S2, show single channels to clearly demonstrate the colocalization/exclusion localization pattern.

5. Line 112-114: Clarify how "actin-depleted membrane curvatures were highly enriched in spectrin and vice versa" suggests "the two scaffolds might aid in shaping negatively curved PM regions"
6. Line 117-119, Fig 1B: Does microcontact printing technique guarantee "non-adhesive surface" zone between fibronectin coated lines? The cells subjected to the technique are still in contact with the plastic and glass surface, to which many cell types show adhesion without fibronectin coating. Cite or show the original work that has tested this. If not shown clearly, the "non-adhesive" cell cortex should be more precisely described throughout the manuscript, for example, focal adhesion-free or integrin-based adhesion-free cell cortex.
7. Line 159-164: Show p-values for statistical analysis of intensity quantifications. Use Standard deviation for error bars. This happens in many plots, which I won't point out every time.
8. Fig 3A-B: Point out actin nodes.
9. Line 178: The statement is counter-intuitive for colocalization analysis. Clarify the limitation.
10. Fig 3E-F: Some kymograph analysis come from a highly compressed depth of samples. Especially, the bottom panels in E and the all F are very difficult to examine and comprehend the complementation and colocalization pattern. In 3E, the cell also seems to be shrinking from one side, which may have partially contributed the regaining of β II-spectrin on the same side. Replacement with another convincing sample without shrinkage would strengthen the argument.
11. Line 211-212, Fig 4A-B: a proper validation for the experiment (to see if spectrin recruitment is dependent on actin or myosin) would be some way to reduce spectrin recruitment (not increasing as seen in the paper). Also, diamide is a general thiol-oxidizing reagent instead of a specific spectrin/PM crosslinker. The image used for diamide experiment in 4A does not appear representative, but rather exaggerated, of the quantification data in 4B (the elevation of global β II-spectrin intensity should be less than 1.3 fold, in fact quite similar to that of the nocodazole experiment).
12. Line 216: Duan et al., 2018 seems irrelevant to the statement.
13. Fig 4E: Statistical analysis missing (p value). Use SD, not SEM for error bar. Remove all the excessive % readings from the figure legends.
14. Line 237: The previous paragraph ended with no significant effect of actin polymerization on cortical spectrin dynamics. Cite previous figures that showed "the spectrin meshwork dynamics depended on contractility and actin polymerization" to clarify the statement.
15. Fig 5A: A few things in the diagram are stylistically inconsistent, hence confusing. For example, does β 2S- Δ ABD construct (green text) lack the PE binding motif (red box)? If not, the red box should be present to be consistent. Also, PE domain alone construct should have the red box in its original place, and the black truncation lines should be used to connect GFP and the red box for stylistic consistency.
16. Fig 5D-E: Use SD, instead of SEM.
17. Fig 5E: show data points. Only mobile fraction data were discussed, leaving out the significance of the half time difference between the constructs. Please explain the half time data.
18. Line 258: "instrumental" does not seem appropriate in the sentence. Also, there is a typo ("protofilaments binding" should be "protofilament binding")

19. Line 268: A typo. "dynamic" should be "dynamics". This typo appears many times in the manuscript.

20. Line 267-268: The conclusion is not well supported due to 1) overall weak functional data, and 2) lack of interpretation of the existing data.

21. Line 385-396, Fig 7G-H: Although the finding seems interesting, the conclusions are overstated solely based on the correlative data.

Reviewer #1 (Remarks to the Author):

This manuscript presents an interesting story of the spatio-temporal dynamics of the spectrin-based membrane skeleton during cell spreading, in relation to that of the better understood actin cytoskeleton. Among the interesting findings are that 1) dynamic assembly of the spectrin-rich membrane domains are mutually exclusive with the F-actin rich domains during cell spreading, thereby coordinating the creation of membrane territories, 2) spectrin-rich domains are associated with regions of negative membrane curvature, 3) spectrin dynamics at the membrane depends on myosin contractility, but not on actin polymerization, and 4) that spectrin-rich domains surround clathrin-coated pits and flow laterally in coordination with the endocytic pits, suggesting the spectrin domain forms a restricting mesh around the forming pits. The response of spectrin-rich domains to whole cell stretching and compression is also discussed. Overall, this study presents novel findings in an area that has not been studied in context of actin dynamics during cell spreading but would be substantially strengthened by experiments to directly test the function of the spectrin-rich membrane domains in cell spreading dynamics and/or in clathrin-mediated endocytosis.

We thank the reviewer for understanding the novelty described by our manuscript. In addressing the criticisms raised, we hope she/he will find the quality improved and acceptable for publication.

In addition, the experiments using cell stretching and compressive stress to investigate how mechanical perturbations affect the spectrin-rich domains are very confusing and not well-related to the higher resolution studies of the relationship of spectrin and actin domain dynamics in the rest of the paper. There are also some interpretations which have been oversimplified and need more clarification and exploration.

We would like to remark that our aim was to provide an integral description of spectrin behavior not only during cell spreading (that purposely serve as a robust model for cell motility/change in shape) but also during perturbation of the cell when already spread, by the means of mechanical perturbations, drugs or microcontact-printing. For these reasons we made use of semi-disposable custom devices which offer alternatives to the standard approaches that require dedicated optical equipment. As the reviewer will see below, we have extensively edited the manuscript as well as presented new experiments that will strengthen our interpretations.

1. Figure 5 describes the dynamics of various spectrin variants (truncations, deletions, specific domains) but does not explore any functional consequences. For example, what is the effect of overexpression of the beta II-spectrin variants missing the actin binding domain or the PE binding domain on rates and morphodynamics of cell spreading and F-actin assembly dynamics? What about the effects on the localization, rates and extents of clathrin-mediated endocytosis?

We made extensive changes to address this criticism. In Figure 6 (please note figure numbers differ from the previous submission) and related movie 6 we provide clearer insights on the unstable behavior of cells overexpressing the variant Δ ABD, and discussed the role of spectrin in creating a continuum between the lamellipodia and the lamella region during cell remodeling.

We now integrate new set of experiments that point toward an altered clathrin pits lifetime in cells overexpressing the variant Δ PE/ANKbs (Figure 8 M). Indeed, this mutant has been renamed since the deletion present does not comprise the sole lipid binding pocket, but spans over the entire Ankyrin binding region. We provided in Figure 6 A the molecular alignment between the spectrin repeats 14-15 of β I and β II-spectrin, where the binding site was originally described (Davis et al. 2009).

From those new set of experiments, we could conclude that actin binding affects cellular morphodynamic properties (Figure 6 and related movies), while membrane/Ankyrin binding is implicated in membrane trafficking mechanisms (Figure 8).

In the discussion, it is proposed that the spectrin creates an interference mechanism to hinder actin polymerization (line 443); this could be tested in the cells overexpressing the spectrin variants missing the ABD.

We thank the reviewer for this suggestion. In our hands, the Δ ABD is correctly localized at the plasmamembrane. From all the experimental evidences collected, we interpret that molecular occupancy of space by spectrin is a condition sufficient to hinder actin polymerization. In particular lamellipodia dynamics, as well as latrunculin A washout, highlighted actin polymerization to occur at spectrin free zones. Therefore, we believe the local ablation of spectrin obtained by mechanical compression represent the most direct experimental evidence of such mechanism, which is now proposed as "potential interference mechanism".

In addition, it is surprising that a spectrin variant missing the ankyrin binding domain, a critical domain for membrane assembly, was not tested. This should be done.

The deletion is now described in Figure 6 A. As pointed in a previous comment, we now have found that membrane binding impairment through the deletion of PE/Ankyrin binding sites has an effect on clathrin pits lifetime (Figure 8 M), but no measurable influence on the morphodynamic properties of the cell during spreading (Figure 6 G and S5). The effects of beta-II spectrin knockdown on actin polymerization dynamics during cell spreading and would also be desirable.

We agree with this comment and tried. However, in light of our disappointing inconsistent data we didn't include those results in the revised manuscript, but we are providing an explanatory figure to the reviewer (see below). Two approaches have been tested: β II-spectrin transient knockdown by shRNA and knockout by Crispr-Cas9 genome editing. In the first case, western blot analysis showed inconsistency in level of β II-spectrin protein content, hampering further cause-effect analysis (3 shRNA tested singularly and in combination, 6 independent experiments). To try to overcome this limitation, we obtained multiple clones by Crispr-Cas9 genome editing fully depleted of β II-spectrin but characterized by high clonal variability and heterogeneity. For example for some clones, β II-spectrin strongly reduced the stability of α II-spectrin counterpart, and re-introducing β II-spectrin in those cells was not sufficient to recover the reduction in cell area observed. Actin and Clathrin analysis showed inconsistent results between different clones as well. At present, we have no confidence in presenting those results that do not fit in the context of this manuscript. We could indeed biasedly "cherry pick" the most suitable clone to fit our results, but we would have no emerging consensus from multiple clone analysis.

In A, among 50 clones 8#, 9#, 10# and 15# were isolated and cultured after receiving Crispr-Cas9; since clone 15# retained β II-spectrin expression, is therefore considered an internal control of the genome editing protocol.

In the 2. Figure 6 presents the effects of whole cell stretching and compression on spectrin assembly at the cell membrane of the ventral surface using a clever apparatus. However, the images presented are mostly of whole cells, so that the relationship of the so-called mesoscale domains of spectrin and F-actin are not apparent; it is very hard to relate this data to that of the previous figures.

We thank the reviewer in finding our apparatus "clever", and we partially disagree with her/his statement. To provide mesoscale details, zooms of nuclear rim formation (Figure 7 C-F and related movie 7) and blebs induced by compression (Figure S6 F-H) are described. Local analysis of cell curvatures and adhesion detachment upon stretching are also documented (Figure 7 A-B, Figure S6 A-C). Regarding the mesoscale analysis, we are also providing an entire new figure (Figure 3), where cells have been spread on microprinted lines to force the formation of negative curvatures. The morphology and dynamics of those specific zones being highlighted by zooms and kymographs.

What does the image time course look like in different cell regions at high magnification? I also wonder what the compression experiment really means when the ventral membrane is being squashed up against the nucleus. This seems like an induced artifact. It would be more informative to test the role of mechanical perturbations at the mesoscale on the different domains shown in Figure 1, for example by AFM or other microscale methods.

We share with the reviewer the concept that the compression experiment is potentially artefactual, such as AFM or micropipette aspiration. Since both techniques have been already reported to describe spectrin and actin behavior by (Duan et al. 2018), we propose here an alternative approach that monitors a different source of external mechanical perturbation. The main strength of our original approach is the reversibility and the high reproducibility allowing cyclical compression-relaxation combined with TIRF imaging at high spatio-temporal resolution. Moreover the “squashing” is not exclusive to the nucleus as we also observed and reported similar behavior induced by large vesicular compartment (see figure S6 D).

More direct information on biomechanical properties is needed before the authors can claim that the “spectrin, actin and plasma membrane create a continuous but dynamic composite material” (line 313). Alternatively, Figure 6 could potentially be left out, and these questions may perhaps be left to a future study, so that the other areas can be better addressed.

We agreed that the “continuity” concept between the players is potentially misleading as they do not form an uninterrupted continuum. The now Figure 7 tackles a topic of broad interest and, as stated in the new paragraph’s title “Spectrin, actin and plasma membrane dynamically interplay during extrinsic perturbations”, the continuity concept has been removed.

3. The authors claim to be studying membrane domains on the order of 100 μm^2 in size, but the FRAP experiments (and the mechanical ones in Figure 5) seem to be on the whole cell or much larger regions. The FRAP experiments should be performed at the appropriate scale (or if they are, this should be more clearly indicated and presented in the figures).

Our FRAP measurements were done in circular (5 μm diameter, roughly 20 μm^2) or squared (25 μm^2) ROIs as stated in the text and in Figure 4 A, not at whole cell. This has been now better emphasized in the text (Line 204-205, legend of Figure 5 and FRAP methods).

With a 100 μm^2 domain, does it matter where the FRAP is performed in the cells with respect to their spreading morphology or at different times during spreading?

Following this comment we performed FRAP analysis in the specific zones of spectrin/actin complementarity described in Figure 1A, showing higher mobility in spectrin-rich cortex compared to cell curvature regions or around stress-fiber rich zones. These data are now presented in Figure 5 B-C and described in the related paragraph. We thank the reviewer for suggesting this experiment which has further the message of differential mesoscale dynamics of the spectrin mesh. We did not performed FRAP during spreading, since ROIs position will be affected by cell movement, also characterized by unpredictable retractions.

The FRAP half-time data in Figure 4D are presented on a cell by cell basis, which reveals the wide variations from cell to cell after the drug treatments. However, the data should also be averaged and statistical analysis performed to evaluate significance of the differences. It is difficult to make a firm conclusion without this. If the changes are not significant perhaps the authors need to rethink their interpretations, and/or test the FRAP in smaller sized membrane domains, as mentioned above.

We thank the reviewer(s) for the suggestion and modify all our frap analysis, now shown as scatter plot representing mean \pm SD of mobile fraction and half time recovery. The mean recovery curves are kept for illustrative purposes. Those data are presented now in Figure 5, confirming our previous conclusions and interpretations.

4. The spectrin-rich domains presumably have actin filaments associated with them; otherwise the spectrin would not form a long-range meshwork (network). The authors refer to these as actin protofilaments; however, assuming this spectrin network is similar to other spectrin networks, they are real actin filaments, just short ones, and should be called “short actin filaments” or “short F-actin”, not protofilaments which implies they are a different structure than normal actin filaments, which is not the case. However, it is worth mentioning that there is no direct evidence regarding the actual actin filament lengths in spectrin networks in any cells besides red blood cells where they are 40 nm long; the authors could make this clear as well..

We thank the reviewer for this comment. Our data highlight the exclusion of spectrin from actin stress fibers, lamellipodia and actin nodes. As suggested, low abundance and isotropic/amorphous actin signal is associated with spectrin in overcontrasted images (now shown in Figure S1 A). We have now implemented the discussion and pointed to spectrin capability to bind only “short-actin filaments” (lines 418-429), and used this terminology throughout the manuscript. Also the lack of direct evidence and structural details of such actin structures except from erythrocytes are highlighted (line 421-423)

Most importantly, the authors should verify that there is F-actin in the spectrin-rich domains by collecting images for a longer time and/or increasing the image intensity (ie, overexposing).

The reviewer will find overcontrasted images in Figure S1 A. Spectrin is excluded from stress fibers while lesser intensity actin is associated with spectrin-rich domain, as previously shown also in Figure S3 B in non-contractile

cortex on the dorsal part of the cell. Those images show how at the back of the lamellipodia spectrin and actin coexist forming beautiful complementary pattern, for either endogenous or over-expressed spectrin.

The other F-actin-rich domains with stress fibers or lamellipodia will need to be blown out to see the lower abundance/density of F-actin in the spectrin-rich domains.

The blown outs of the mentioned domains are now shown in Figure 1A.

This reminds me of the situation when studying F-actin in the cell nucleus; perhaps those approaches could provide some helpful tips here. Assuming the authors can obtain the sensitivity to visualize the F-actin in the spectrin-rich domains, it would be very interesting to perform FRAP and compare this F-actin dynamics to the spectrin dynamics and to F-actin dynamics in the other types of domains. Their observation in Figure 4 that LatA treatment did not affect the spectrin dynamics assayed by FRAP is consistent with the idea that the short actin filaments in the spectrin network are stable; as shown previously for RBC actin filaments in the red cell spectrin-based membrane skeleton by Gokhin et al., *MBoC* 26:1699, 2015.

We are pleased to have reached the same conclusion: short actin filament and the spectrin meshwork associated with them are indeed stable structures and we included this citation in the discussion of the Latrunculin A treatment (lines 421-423). However, not only we have the problem of the sensitivity required to visualize low abundance actin in non-contractile cortex as shown for cells fixed in early spreading (Figure S3 A-B), as pointed by the reviewer, but also we cannot be sure about its direct association with spectrin. We may just FRAP cortical actin and not spectrin-bound actin, which has been already reported by many others.

5. The association of the spectrin-rich domains with the regions of negative membrane curvature shown in Figure 6 is interesting but should be better documented and quantified in the context of the spreading experiments— ie, what about negative curvature regions in spreading cells? Crawling cells? etc. In addition, this should be discussed in the context of previous studies of actin and myosin association with negative or positive membrane curvatures. For example, Elliott, Fischer et al. *Nat Cell Biol.* 17:137, 2015. Lou, Zhao et al. *PNAS* 116 (46):23143, 2019, among others. Following reviewer' suggestion, we now document in a new Figure 3 and related movie 2 the formation of negative curvatures by spreading cells on microprinted patterns, specifically fibronectin-coated lines separated by 12-16 μ m of non-adherent surface. As already quantified in Figure 2 D-G, spectrin accumulate in retracting curvatures, while actin bridges marked protruding curvatures as we previously showed (Rossier et al., 2010). Micrometer-scale curvatures described by (Elliott, Fischer et al. 2015) and (Lou, Zhao et al. 2019) are extremely interesting, but on smaller scale than the curvatures described in our manuscript.

6. The spectrin-rich domains should be referred to as the spectrin-based membrane skeleton, or spectrin-rich domains, not the spectrin cytoskeleton; the cytoskeleton extends into the cell whereas the key feature of the spectrin-based membrane skeleton is that it is associated with the membrane and not extending into the cytoplasm. The authors also use "membrane-associated spectrin meshwork" on line 207-this is fine too.

We removed "spectrin cytoskeleton" throughout the text and comply with the nomenclature suggested by the reviewer.

7. The authors conclude from several of their experiments that the spectrin meshwork dynamics confers resilience to the cell; e.g., see line 268. However, the resilience of the spectrin meshwork is not measured directly, for example, using AFM or micropipette pulling or in poking experiments at the mesoscale; see also comment #2 above.

We agree with this comment and recognize the limits of our work. The speculations about the resilience have been removed

8. The term 'mesoscale' is not clearly defined at the beginning of the manuscript.

We define mesoscale as the intermediate scale between molecular scale and the cell scale. Given the length of individual tetramers (80 to 200nm), for spectrin this range between 1 and 10 μ m² (lines 33-35). This concept is also highlighted by the sentence "spectrin works as a sum larger than its individual parts (dimers and tetramers)" in the discussion (lines 445-452).

9. line 37. Spectrin is not ubiquitous. It is only in metazoan cells.

We corrected the sentence: "This protein is expressed in all metazoan cells.'. The abstract has been modified accordingly.

10. lines 64-65. The original electron microscopy studies from Jeannine Ursitti should be mentioned in discussion of the lengths for the spectrin tetramers in situ in the erythrocyte membrane skeleton Ursitti et al., 1991 *Cell Motil Cytoskeleton* 19(4):227; Ursitti and Wade, 1993 *Cell Motil Cytoskeleton* 25(1):30. It might also be helpful for the authors to read the reviews on the RBC spectrin membrane skeleton to improve the context of the introduction and

discussion, especially in terms of the role of spectrin in mechanoprotection (lines 70-71); e.g. from Mohandas and Evans, 1994, *Ann Rev Biophys Biomol Struct* 23:787, and Mohandas and Gallagher, 2008 *Blood* 112:3939.

These citations have been added to the introduction

11. The Lardennois reference is not correct. The Frick, Schmidt and Nichols reference is missing. There may be other errors in the reference list.

We thank the reviewer, all the references have now been checked

12. The antibody binding sites (epitopes) of the anti-spectrin antibodies should be stated in the results and in the methods, and their locations taken into account in the results and discussions. Some antibodies recognize the middle of the spectrin tetramer and others the ends near the actin binding site. It is conceivable that localization results may differ depending on the location of the antibody binding. Also, the locations of the GFP tags should be stated.

A scheme of the spectrin dimer, localization of the GFP-tag and the different epitopes recognized by the antibodies implemented in this study has been added to Figure S1 B. The location of the GFP tagged is also shown in Figure 6A

Reviewer #2 (Remarks to the Author):

I am reviewing the manuscript entitled 'Mesoscale Dynamics of Spectrin and Acto-Myosin shape Membrane Territories during Mechanoresponse' submitted to Nat Comm by Ghisleni, Gauthier and co-authors. The manuscript provides an in-depth characterization of the interplay between actin and spectrin cytoskeletons and uncovers several novel and unexpected features of the cortical cytoskeleton and thus complements existing, earlier work from Bennet et al. The authors show a seemingly contradictory and antiphasic role of actin and spectrin cytoskeleton during the cell spreading process but also during the response of the cell to external mechanical stimuli. They also investigate the spectrin dynamics as being dependent on myosin contractility, which requires the actin binding domain. At the same time, spectrin occupies territories distinct from actin, a fact that seemed to evade my intuitive understanding. The authors also investigate the dynamics of spectrin cytoskeleton under compression and show a striking exclusion of spectrin from the perinuclear actin belt and thus might contribute to our understanding of the cellular mechanoresponse during confined cell migration. The present topic is timely, the data is well analysed and described extremely well, and the manuscript is written clearly with display items of superior quality. I thus think the article is suitable for publication after minor revisions.

We are pleased for this enthusiastic commentary and appreciation of our work, for which we thank the reviewer.

General comment:

Spectrin and actin have distinct domain during expansion and retraction and during mechanical compression. This is particularly surprising by the fact that deletion of the actin binding domain changes the properties of the spectrin network. How does spectrin do this and what is the role of the actin binding domain, if it does not colocalize with actin? Perhaps I missed that point in the manuscript but the understanding might benefit from an explicit paragraph in the discussion.

In the Result ("Spectrin and Actin define distinct but complementary plasma membrane territories in cells of different origins") and Discussion we now clarified this point (lines 418-429). The spectrin-based membrane skeleton comprise only short actin filaments (similar to ≈ 40 nm filaments described in erythrocytes); those isotropic filaments, if visible, may only be detected in the new overcontrasted images we provide (Figure S1 A) and will be bound through the actin binding domain. We could speculate such structures share homology with those found in erythrocytes, albeit direct structural evidences are still lacking. The spectrin meshwork is however excluded from anisotropic/morphodynamic actin structures such as stress fibers, lamellipodia, filopodia, blebs and nodes.

In Figure 6, no details on the cell stretching experiment have been giving, e.g. How long was the stretch applied for and how was the strain field quantified? I assume that cell at the center of the membrane experience more stretch than cell on the periphery? Do the cells experience the stress delivered biaxial or radial? Is the device attached to a previous publication? If not, more details are mandatory before publication.

We provide those details in the related Result' (lines 280-328) and Methods' paragraphs (lines 1005-1063) to fulfill the request of the reviewer. Please also refer to supplementary figure S8 where the reviewer can find the exploded-view drawings of both devices implemented in the study. Specifically in the result's section: "we imposed a sequential 5%-step size increase in stretch, up to 30% of the initial area; at each step a single frame was recorded to document cell and proteins behavior in both fluorescent channels". Further requested details are provided in the Method's section, as well as in the Figure S8.

Have the authors tried to knockdown the endogenous β -spectrin in the cells?

We include for reviewing purpose only, the flaw and inconsistencies in the analysis of knock-down and knock-out β II-spectrin in MEFs. We didn't include those results in the revised manuscript but we are providing an explanatory figure to the reviewer. Two approaches have been tested: β II-spectrin transient knockdown by shRNA and knockout by Crispr-Cas9 genome editing. In the first case, western blot analysis showed inconsistency in level of β II-spectrin protein content, hampering further cause-effect analysis (3 shRNA tested singularly and in combination, 6 independent experiments). To try to overcome this limitation, we obtained multiple clones by Crispr-Cas9 genome editing fully depleted of β II-spectrin but characterized by high clonal variability and heterogeneity. For example for some clones, β II-spectrin strongly reduced the stability of α II-spectrin counterpart, and re-introducing β II-spectrin in those cells was not sufficient to recover the reduction in cell area observed. Actin and Clathrin analysis showed inconsistent results between different clones as well. At present, we have no confidence in presenting those results that do not fit in the context of this manuscript. We could indeed biasedly "cherry pick" the most suitable clone to fit our results, but we would have no emerging consensus from multiple clone analysis.

MEFs *sptbn1* KO (CRISPR/Cas9)

A) Western blot of different clones B) Cell area measurements
C) Whole-cell fluorescence intensities for different markers

In A, among 50 clones 8#, 9#, 10# and 15# were isolated and cultured after receiving Crispr-Cas9; since clone 15# retained βII-spectrin expression, is therefore considered an internal control of the genome editing protocol.

The recovery seems to be very fast in comparison to other literature values, so I might conclude the the labeled spectrin is predominantly excluded from the native network. Can the authors comment on this?

Our FRAP data are indeed faster than similar analysis in neurons (Zhong, He et al. 2014). However, we found the recovery in line with the results obtained by (Duan, Kim et al. 2018), in terms of mobile fraction (which is directly linked to the inclusion in the native meshwork) as well as the highly variable half time recovery (which is more dependent on molecular binding events). These results are now discussed and interpreted in the paragraph “Spectrin mobility is influenced by cortex topology and depends on contractility”. We also performed a zonal FRAP assay showing differential mobility of the meshwork in the different zones of actin/spectrin complementarity introduced in figure 1, which also address the point raised (see new Figure 5 A-C). This new experiment showed higher mobility in spectrin-rich cortex compared to cell curvature regions or around stress-fiber rich zones, which further the message of differential mesoscale dynamics of the spectrin mesh.

Also, apart of the PH domain, b-spectrin has an ankyrin binding domain - do the authors expect a role for this?

The deletion of PE domain is in fact a ΔPE/Ankyrin binding sites deletion and is now described better in Figure 6 A. New set of results suggest membrane/Ankyrin binding impairment has an effect on clathrin pits lifetime (Figure 8 M), but not on morphodynamic properties of the cell.

1) Line 37: the key component spectrin; spectrin is not a single component and neither is it a single protein. “a key component” was replaced by “an important element”.

2) Line 67: Brown et al, PLOS Comp Biol, 2015 could be cited here

The citation has been added

3) Line 73: instead of mechanical tension, spectrin seems to protect axons from mechanical deformation during buckling by keeping them under tension

“Mechanical tension” has been modified to “protect axons from mechanical deformation by keeping them under constant tension”, according to the reviewer’ comment

4) Line 212: diamide is not a common drug. Can the authors quickly introduce what it does and why it is specific for spectrin?

A sentence has been added in the description of the related experiments: "a drug that targets sulphhydryl groups and induces spectrin recoiling into ring-like structures". Briefly, diamide is a drug with poor membrane permeability able to modify sulphhydryl groups. It therefore affects membrane associated proteins like spectrin, specifically its ability to bind protein 4.1 and to induce recoiling into ring-like structures (Becker, Cohen et al. 1986). However, it is not specific for spectrin but has been applied here to provide an extreme negative control for spectrin dynamics perturbation.

5) Line 215: The result suggests that microtubules antagonize spectrin assembly at the plasma-membrane. Can the authors discuss this?

We discussed Nocodazole effect in lines 225-230. Briefly, Jenkins and Bennett extensively analyze this effect on a different timescale (up to 3h). They point to a MT-dependent organization of Ankyrin-G domains, which inevitably has consequence on spectrin. Albeit we agreed on this observation, our goal was to observe an effect in a much shorter time-scale (5' to 30'). The interpretation of the reviewer is indeed correct and 30' of Nocodazole treatment affect spectrin organization at the PM; however, in a shorter timescale (5') MT destabilization does not produce an immediate effect, while depolymerization at this time is effective (see our FRAP results in Figure 5 E). By contrast, at this short time scale, effects were evident when myosin activity was blocked or actin filament stabilized (see particularly new results by jasplakinolide treatment in Figure 5 and S4).

6) Figure 7 could be split and panel I deserves it own figure.

We agree with the reviewer's suggestion. The model is now presented alone in Figure 9

7) several typos throughout methods including the following ...

8) Line 972, monitorable = detectable

9) Line 1037: Graphpad reads Graphdap

10) Figure S3 panel C, Axis label reads Intgrated instead if Integrated

7-10: We thank the reviewer for spotting these. Changes and typos errors have been corrected

11) The actin signal in the bleb is due to polymerized actin or diffusion of RFP-actin into the bleb?

With our resolution, we observed mostly what looks like diffusion of soluble RFP-actin into the "mechanically-induced" blebs. Indeed no clear "near membrane reinforcement" of structured fluorescence suggesting polymerization were observable in the first frames after bleb appearance. However, we observed actin fluorescence strengthening in later time frames suggesting condensation-polymerization during the resorption phase as highlighted in our Figure S6 F-G. This is described at lines 329-331.

Reviewer #3 (Remarks to the Author):

Summary:

In the manuscript entitled “Mesoscale dynamics of Spectrin and Acto-myosin shape membrane territories during mechanoresponse”, Ghisleni et al. find that spectin and acto-myosin cytoskeletons are spatially distinct but cooperate during cell mechanoresponses specifically cell adhesion and compression. To this reviewer’s knowledge, this work is the first detailed demonstration of the dynamic interplay between the acto-myosin and spectrin cytoskeletons at the cell cortex. This reviewer found the writing, experimental creativity/design, and high-quality data garnered from the techniques used in this manuscript to be compelling. Further, while much attention has been focused on cytoskeletal interconnectedness between the actin and microtubule cytoskeletons exactly how these dynamics connects to the complicated interactions with lipids, vesicles, and the cell cortex has been vastly understudied. This work clearly advances the cytoskeletal crosstalk and mechanobiology fields - congratulations to the authors on this **outstanding** work.

We thank the reviewer for sharing with us her/his enthusiasm on our findings and qualifying our work as outstanding!

Below are specific **minor** concerns and points of clarification:

- The complimentary pattern between Spectrin and actin in multiple human and mouse cell lines is really interesting. Mention/speculation on what molecules/mechanisms may be directly mediating this coordination in the discussion could make this work more accessible to a broader readership. Further, given the specific regions of complimentary patterning (leading edge, stress fibers, negative membrane curvature) I am curious if septins or BAR proteins mediate/contribute to such interactions.

In reply to other reviewers, we extended our discussion (lines 418-429) and provide potential speculative mechanisms that generate this complementary pattern. Focusing on reviewer’ curiosity, a very recent publication (Hamdan et al., 2020) identified a physical link between Ankyrin/Spectrin scaffolds and septins by biotin ligase approach. We agree with the reviewer about the benefits and we included this citation in the discussion, specifically at support of the sentence “This level of specificity likely requires the use of accessory proteins”. However, such analysis is restricted to the axon initial segment and any further parallelism with our study is highly speculative at this stage

- Many groups have started using spectrin “seeds” to assess aspects of actin dynamics in microfluidic systems. Could the complimentary pattern be mimicked in an in vitro system? Would actin polymerization be in distinct non-spectrin regions similar to the observations in cells? These experiments would likely be very challenging and require reconstituted lipids, making this suggestion well beyond the scope of this work.

We thank the reviewer for the interesting idea, which has “seeded” many potential future perspectives. Unfortunately, at present in vitro experiments are beyond our technical capabilities.

- The drug treatments employed throughout this work are extremely helpful in dissecting the role of spectrin-acto-myosin crosstalk, but they are also kind of like hitting the cell with a hammer. If the data is available, siRNA knockdown of a myosin phenocopying the blebbistatin data or additional text in the manuscript about the limitations of this broad-spectrum drug could be helpful. In this similar vein, are specific components of actin assembly i.e. Arp2/3 complex (also critical for endocytosis) is regulating the actin behaviors described here. Have the authors considered using CK666/Arpin to assess the actin assembly factors possibly involved with this crosstalk?

We shared the reviewer considerations on the pitfall of drug treatment approaches. For this reason, we monitored the acute phase of drug treatment (5’ to 30’) instead of simply comparing long-term effects in parallelized analysis. Also, FRAP dynamic studies were performed on the same cell before and after drug administration to precisely detect the acute effect. We have not consider CK666/Arpin but we have added another treatment to stabilize the actin by jaspakinolide. This can now be found in Figure 5 and S4. Jaspakinolide has a strong effect indicating that the stabilization of F-actin increase spectrin signal at the basal membrane by reducing the mobility of the spectrin meshwork (measured by FRAP).

- While there are hundreds of endocytosis events happening at any given time during image acquisition for experiments described in Figure 7, quantification of BII-S “holes” and CHC/actin/AP2a puncta could give the reader a better idea of the frequency of these associations.

To comply with the mesoscale description, in the revised version of this manuscript we quantified pits density in spectrin-rich and spectrin-poor domains, as well as the pits lifetime at these different zones. Reviewers can find these new data in figure 8 I-L. Overall pits form less frequently and have longer lifetime at regions of high spectrin density,

suggesting the existence of a molecular occupancy mechanism that partially impede vesicle fusion at spectrin-dense zones.

Minor formatting considerations:

We thank the reviewer for spotting these typos, that we corrected

- Line 486 a space is needed between plasma and membrane.
- Line 455 The word “figure” in the Figure 7I callout should be capitalized for consistency.
- Line 462, while this reviewer appreciated the “ménage a 4” callout due to similar actin-based mechanisms, perhaps a more specific way for describing the Spectrin convergence point between acto-myosin contraction, polymerization, and exo/endocytosis is more appropriate.

We thank the reviewer suggestion, but we would like to keep the sentence “by an orchestrated “menage a 4” between PM, spectrin, exo/endocytosis and acto-myosin contraction-polymerization”

Reviewer #4 (Remarks to the Author):

Ghisleni et al. examined the mesoscale dynamics of cortical actin and spectrin cytoskeleton during cell cortex remodeling. The strengths of the paper include the discovery of mutually-exclusive cortical zones of actin and spectrin networks, a range of biophysical assays to show the mechano-responsive behavior of actin and spectrin networks, and quantitative image analysis throughout the manuscript. Taken together, the authors concluded that the antagonistic interplay between cortical actin and spectrin networks is required for organizing the lipid bilayer into spatially confined cortical territories during cell mechanoresponse.

We would like to thank the reviewer for carefully and critically reading our manuscript. We are pleased the reviewer values the strengths of our paper, and hope to have solved the concerns raised in this revised version

Major concerns:

1. Although the observations of the complementary localization of actin and spectrin are convincing and interesting, the manuscript is largely descriptive without providing significant mechanistic insight. For example, what triggers the mutually exclusive localization of actin and spectrin?

All the data collected in this manuscript point to the exclusion of spectrin from actin structures linked to morphodynamic regulation, such as lamellipodia, stress fibers and actin nodes. Even in the cortex spectrin is distinct from cortical actin as shown in Figure S3 A-B. From red blood cell evidence spectrin only associate with short actin filaments (potentially visible in overcontrasted images now shown in Figure S1 A), and the effect of Actin Binding Domain deletion mutant expression is pointing toward a clear and functional actin-spectrin interaction. This also explain the coordinated motion observed. The exclusion, as the reviewer highlighted, is much harder to explain but we discussed this point in several places in the revised manuscript, in particular at lines 418-429.

Briefly, our results show that spectrin and plasmamembrane are associated at constant whole-cell ratio. However, actin dynamics is constantly affecting this ratio at mesoscale level, like at lamellipodia-cleared spectrin, or on the contrary at actin nodes-driven spectrin enrichment. This is consistent with a local occupancy mechanisms, which is supported by our compression experiment (see next answer to the reviewer)

Does spectrin exhibit actin polymerization- and/or actomyosin-induced mechanosensory response?

We believe spectrin does not possess actin polymerization capacity. On contrary, our compression experiments showed that spectrin clearance triggered actin polymerization (Figure 7 C-F), which point to a “potential interference mechanism” as mentioned in the discussion. To our knowledge this represents an unprecedented mechanism in the cortex dynamics. Albeit a direct link remains to be found, we cannot exclude a molecular sensing mechanism led by spectrin, which will be speculative in the context of this paper. It is also interesting to note, that both have affinity for similar phosphoinositides (as pointed in our previous findings in macrophages Masters et al., 2016). This is also added in the discussion (lines 427-429).

How contractility may regulate spectrin recruitment?

We quantified spectrin deposition during spreading and found linear correlation with cell area (Figure S3 C-D), suggesting recruitment is mainly led by membrane binding. Therefore contractility does not regulate spectrin recruitment in our system but is involved in its reorganization. We worked through the text to avoid this incorrect interpretation.

Also, how are the AP2 pits hooked to the spectrin network?

We erroneously used the term “hooked” (now corrected with the most appropriate terminology “fenced”) and we thank the reviewer for pointing this out. As show in Figure 8, pits occupy the open space of the spectrin mesh and flow in coordination when the meshwork moves, in compliance with the fluid mosaic model of plasmamembrane organization and the molecular occupancy mechanism that lowered pits density at spectrin-rich domains (as shown by the new experiments in Figure 8 I-L).

2. It is difficult to follow the biophysical experiments (Line 269-359, Fig 6, Fig S5) for several reasons: 1) poor description of the experiments and lack of interpretation of the results, 2) inconsistent order of experiments between the main text and the figures, and 3) mislabeling of the in-text citations, figures and figure legends.

We thank the reviewer for these comments. The entire paragraph has now been edited (lines 280-336), text and figures are now following the same order (Figure 7 and S6). More details have been provided also in the Method's section regarding the experimental design and the two devices. We hope all the criticisms have now been fixed in the revised manuscript.

3. More experiments are required to strengthen some conclusions. For example, the investigation of the effect exerted by the actin cytoskeleton on the spectrin network is based on a single pharmacological perturbation

experiment. This should be further strengthened with additional actin depolymerizing and stabilizing reagents to disrupt the actin network for more thorough analyses.

We thank the reviewer for the criticism. We extensively edited Figure 5 and Figure S4, where the F-actin stabilizer jasplakinolide complement the other pharmacological perturbations. As the reviewer can observe, jasplakinolide has a strong effect indicating that the stabilization of F-actin increase spectrin signal at the basal membrane by reducing the mobility of the spectrin meshwork (measured by FRAP). Concerning the single pharmacological perturbation during spreading, myosinII inhibition by blebbistatin is the only effective compound that does not impede cell engagement with the substrate and initial spreading, while actin- or microtubule-directed perturbations completely prevent the isotropic cell spreading analyzed. This is consistent throughout the literature and a known common observations among independent groups.

4. Despite extensive quantification in the manuscript, statistical analyses are often lacking.

We now consistently show data as mean±SD, and the opportune statistical analyses have now been performed. Please find all the detail in figure legends.

Specific comment:

1. The title: The article is mainly about the mutually exclusive actin and spectrin zones (and endocytosis), hence the title needs to be revised to reflect the main findings of the work.

Following reviewer's comment, title has been revised to reflect the "Complementary Mesoscale Dynamics of Spectrin and Acto-Myosin shape Membrane Territories during Mechanoresponse". Exclusivity of actin and spectrin zones will represent an overstatement, while endocytosis is implicit in the definition of membrane territories.

2. Line 34: Show clear definition of "mesoscale" in the introduction. (100nm~10um)

In the introduction, we now define "mesoscale" of non-polarized meshwork as the intermediate scale between molecular scale and the cell scale. Given the length of individual spectrin tetramers (80 to 200nm), for meshwork under study this range between ≈1 and 10μm² (lines 33-35). This concept is also highlighted by the sentence "spectrin works as a sum larger than its individual parts (dimers and tetramers)" in the discussion.

3. Line 56: "triangular-like" should be "triangle-like"

This has been corrected

4. For Fig. 1A inboxes on the right, Fig S2, show single channels to clearly demonstrate the colocalization/exclusion localization pattern.

Single channel images and blown outs are now shown in Figure 1 A and S2 A.

5. Line 112-114: Clarify how "actin-depleted membrane curvatures were highly enriched in spectrin and vice versa" suggests "the two scaffolds might aid in shaping negatively curved PM regions"

To address this criticism we performed new spreading experiments on fibronectin-coated lines interleaved with non-adhesive substrate, in order to force cells creating negative curvatures. Also in this context the duality between actin-rich and spectrin-rich curvature was highlighted (new Figure 3 and related movie) Briefly spectrin-rich curvatures occur in retracting curvatures (Figure 3 A for spreading cell, C for polarizing cell), while protruding negative curvatures are sustained by a leading actin arc that we previously described in details (Rossier et al., 2010), followed by the spectrin meshwork (Figure 3 E in already spread cell). Moreover, zonal FRAP assay was also performed, indicating different dynamics at curvature regions compared to spectrin-rich cortex (Figure 4 A-C). Results are described at lines 203-211.

6. Line 117-119, Fig 1B: Does microcontact printing technique guarantee "non-adhesive surface" zone between fibronectin coated lines? The cells subjected to the technique are still in contact with the plastic and glass surface, to which many cell types show adhesion without fibronectin coating. Cite or show the original work that has tested this. If not shown clearly, the "non-adhesive" cell cortex should be more precisely described throughout the manuscript, for example, focal adhesion-free or integrin-based adhesion-free cell cortex.

Following the reviewer comment, a clarification is needed. The results and methods' sections have been implemented accordingly. Specifically, microcontact printing provides patterned adhesive zones (coated by fibronectin), while the following passivation step by PLL-PEG ensure the non-adhesiveness of the non-patterned surface. This is an established approach in the lab (Monzo et al. 2016, Pontes et al. 2017) that we pioneered more than 10 years ago (Rossier et al. 2010).

7. Line 159-164: Show p-values for statistical analysis of intensity quantifications. Use Standard deviation for error bars. This happens in many plots, which I won't point out every time.

We have fixed this for almost all quantifications (see also the other points raised by the reviewer).

In the particular case pointed here by the reviewer we want to stay with SE as the deviation showed by SD is too big due to the extremely large data points considered (shown in Figure S3 E), and unnecessarily weaken the data presented.

The representation is peculiar and complex, and is a way that we chose to average all the movies reflecting cells behavior as singularly show on the panels D and F. The main message presented is the cross trend with more actin fluorescence in protruding zones and more spectrin fluorescence in retracting zones. The kymographs are very clear by themselves. Each cell averaged here is a true independent experiment. Thousands of points are averaged for each cell to generate a single graph for each fluorescent marker, then those graphs are averaged again together. As such, a single cell is a true independent experiment and SE is most appropriate with 9 cells for control and 5 for Blebbistatin. From cell to cell, the trend is solid and it is reflected all over the manuscript (actin leads protrusion, spectrin is more intense in retraction) and we don't feel the need here to change SE for SD.

We hope this is not a blocking point for the reviewer otherwise we will change our representation for something that the reviewer may find more appropriate.

8. Fig 3A-B: Point out actin nodes.

Actin nodes are pointed out in all the kymographs in Figure 4

9. Line 178: The statement is counter-intuitive for colocalization analysis. Clarify the limitation.

We clarify with "no colocalization with those actin structures"

10. Fig 3E-F: Some kymograph analysis come from a highly compressed depth of samples. Especially, the bottom panels in E and the all F are very difficult to examine and comprehend the complementation and colocalization pattern. In 3E, the cell also seems to be shrinking from one side, which may have partially contributed the regaining of β II-spectrin on the same side. Replacement with another convincing sample without shrinkage would strengthen the argument.

We commonly disagree with the reviewer on this point. The shrinkage effect occurred upon the washout of the drug and is a common observation upon latrunculin A treatment. Remarkably, during drug administration the collapsing structure is actin and not spectrin, while spectrin condensation is specifically driven by the formation of actin nodes during washout. We included the movie of the entire cell and not only the zooms presented in Figure 3 for a more comprehensive analysis of Latrunculin A effect on both structures (see Movie 4). The reviewer might also note how these experiments complement the endogenous and unperturbed observation of actin nodes formation during spreading, and are intended to sustain the claim that nodes coalescence drives spectrin reorganization. Nodes formation can also be found on our early washout experiments on lines presented in the Figure S3 F, where actin nodes are nucleated in the openings of the endogenous spectrin meshwork, particularly visible above the non-adhesive substrate.

11. Line 211-212, Fig 4A-B: a proper validation for the experiment (to see if spectrin recruitment is dependent on actin or myosin) would be some way to reduce spectrin recruitment (not increasing as seen in the paper). Also, diamide is a general thiol-oxidizing reagent instead of a specific spectrin/PM crosslinker. The image used for diamide experiment in 4A does not appear representative, but rather exaggerated, of the quantification data in 4B (the elevation of global β II-spectrin intensity should be less than 1.3 fold, in fact quite similar to that of the nocodazole experiment).

We thank the reviewer for the suggestion and extensively edited this paragraph and related Figure 5 and S4. We complemented this set of experiments with an F-actin stabilizer (Jasplakinolide) and found similar increase in spectrin fluorescence and reduced mobility (by FRAP). At present, none of the treatments showed the capability to reduce spectrin recruitment or association with the plasmamembrane. Our interpretation, is that spectrin recruitment is mainly led by the association with the plasmamembrane (as quantified in Figure S3 C), while acto-myosin stabilization impair spectrin re-organization. A better description of the diamide effect is also provided (lines 216-218). The panel in the old Figure 4 A-B has now been moved to the new Figure S4 A-B, and more representative images of the effects observed are shown.

12. Line 216: Duan et al., 2018 seems irrelevant to the statement.

We removed the reference

13. Fig 4E: Statistical analysis missing (p value). Use SD, not SEM for error bar. Remove all the excessive % readings from the figure legends.

The figure has been extensively edited (now Figure 5), all graphs are now showing SD and p values.

14. Line 237: The previous paragraph ended with no significant effect of actin polymerization on cortical spectrin dynamics. Cite previous figures that showed “the spectrin meshwork dynamics depended on contractility and actin polymerization” to clarify the statement.

The entire paragraph has been edited to describe an entire new set of data. The claim on “actin polymerization” has been removed to comply with the effect observed upon the new experiments using jasplakinolide treatment.

15. Fig 5A: A few things in the diagram are stylistically inconsistent, hence confusing. For example, does β 2S- Δ ABD construct (green text) lack the PE binding motif (red box)? If not, the red box should be present to be consistent. Also, PE domain alone construct should have the red box in its original place, and the black truncation lines should be used to connect GFP and the red box for stylistic consistency.

We thank the reviewer for pointing this out, the diagram has now been corrected

16. Fig 5D-E: Use SD, instead of SEM.

All graphs have been corrected accordingly

17. Fig 5E: show data points. Only mobile fraction data were discussed, leaving out the significance of the half time difference between the constructs. Please explain the half time data.

All FRAP graphs have now been edited according to reviewer’s comment. Half time data are indeed highly variable and not statistically different (as they were reported by Duan et al., 2018). This is now discussed in the text, where we also provide interpretation of this results: “Half-time recovery rates were overall highly variable, potentially influenced by many protein binding events independent from actin that could not be determined here” (lines 209-211)

18. Line 258: “instrumental” does not seem appropriate in the sentence. Also, there is a typo (“protofilaments binding” should be “protofilament binding”)

The term protofilaments is now removed, replaced by “short actin filaments”. We agree with the reviewer and removed also the term “Instrumental”

19. Line 268: A typo. “dynamic” should be “dynamics”. This typo appears many times in the manuscript.

The entire manuscript has been proof-read and we hope we have corrected most if not all errors of this type.

20. Line 267-268: The conclusion is not well supported due to 1) overall weak functional data, and 2) lack of interpretation of the existing data.

We changed our conclusion and in particular we removed the sentence “conferring resilience to the cell”, which is indeed not supported by functional data (see lines 278).

21. Line 385-396, Fig 7G-H: Although the finding seems interesting, the conclusions are overstated solely based on the correlative data.

This criticism made us conceive the experiment now shown in Figure 8 I-L, where focusing on low- and high-spectrin density domains we identified pits to form more frequently at low-density zones. Moreover, those pits were resolved faster (in terms of lifetime) than pits formed at high-spectrin density zones. Altogether, “These results support a fencing mechanism by spectrin condensation that affects local endocytic capacities”, as now stated in the related result’ paragraph (lines 393-398).

References

- Becker, P. S., C. M. Cohen and S. E. Lux (1986). "The effect of mild diamide oxidation on the structure and function of human erythrocyte spectrin." *J Biol Chem* **261**(10): 4620-4628.
- Davis, L., K. Abdi, M. Machius, C. Brautigam, D. R. Tomchick, V. Bennett and P. Michaely (2009). "Localization and Structure of the Ankyrin-binding Site on β 2-Spectrin."
- Duan, R., J. H. Kim, K. Shilagardi, E. S. Schifffhauer, D. M. Lee, S. Son, S. Li, C. Thomas, T. Luo, D. A. Fletcher, D. N. Robinson and E. H. Chen (2018). "Spectrin is a mechanoresponsive protein shaping fusogenic synapse architecture during myoblast fusion." *Nature Cell Biology* **20**(6): 688.
- Elliott, H., R. S. Fischer, K. A. Myers, R. A. Desai, L. Gao, C. S. Chen, R. S. Adelstein, C. M. Waterman and G. Danuser (2015). "Myosin II controls cellular branching morphogenesis and migration in three dimensions by minimizing cell-surface curvature." *Nat Cell Biol* **17**(2): 137-147.
- Lou, H. Y., W. Zhao, X. Li, L. Duan, A. Powers, M. Akamatsu, F. Santoro, A. F. McGuire, Y. Cui, D. G. Drubin and B. Cui (2019). "Membrane curvature underlies actin reorganization in response to nanoscale surface topography." *Proc Natl Acad Sci U S A* **116**(46): 23143-23151.
- Masters, T. A., M. P. Sheetz and N. C. Gauthier (2016). "F-actin waves, actin cortex disassembly and focal exocytosis driven by actin-phosphoinositide positive feedback." *Cytoskeleton (Hoboken)* **73**(4): 180-196.
- Monzo, P., Y. K. Chong, C. Guetta-Terrier, A. Krishnasamy, S. R. Sathe, E. K. Yim, W. H. Ng, B. T. Ang, C. Tang, B. Ladoux, N. C. Gauthier and M. P. Sheetz (2016). "Mechanical confinement triggers glioma linear migration dependent on formin FHOD3." *Mol Biol Cell* **27**(8): 1246-1261.
- OM, R., G. N, B. N, V. W, F. MA, A. P, H. ER, M. A, G. S, K. MS, H. JC and S. MP (2010). "Force Generated by Actomyosin Contraction Builds Bridges Between Adhesive Contacts." *The EMBO journal* **29**(6).
- Pontes, B., P. Monzo, L. Gole, A. L. Le Roux, A. J. Kosmalska, Z. Y. Tam, W. Luo, S. Kan, V. Viasnoff, P. Roca-Cusachs, L. Tucker-Kellogg and N. C. Gauthier (2017). "Membrane tension controls adhesion positioning at the leading edge of cells." *J Cell Biol.* , 216(9), pp. 2959-2977
- Zhong, G., J. He, R. Zhou, D. Lorenzo, H. P. Babcock, V. Bennett and X. Zhuang (2014). Developmental mechanism of the periodic membrane skeleton in axons. *eLife*. **3**.

REVIEWERS' COMMENTS:

Reviewer #1 (Remarks to the Author):

This revised manuscript further improves and substantiates the authors' novel findings regarding the complementary dynamics and locations of the actin-rich cortex with respect to the spectrin-based membrane skeleton during cell mechanoresponses. All of my general and specific concerns have been addressed with new experiments, new analyses, as well as clarification and improved interpretations. The new experiments using the micropatterned substrates to demonstrate the complementary domains are excellent, as are the new experiments to test the functions of β II spectrin domains in endocytosis. The inclusion of statistical analyses has also substantially improved the confidence in the data, and interpretations and conclusions have been clarified. The concept that the spectrin network responds to mechanical forces by condensation rather than by molecular extension or coiling of individual spectrin molecules is novel. Overall, this study presents exciting and novel findings about spectrin-based membrane skeleton interactions and coordination with actin cytoskeleton dynamics during cell spreading and mechanoresponses. In particular, the common belief has been that the spectrin-based membrane skeleton is only important for highly specialized cells such as neurons and erythrocytes, but this study now shows a critical role for the spectrin-based membrane skeleton in coordinating actin dynamics in so-called 'typical' cultured cell types that are studied by many labs. This will need to be accounted for in future studies.

I have a few minor suggestions for improvements in the text, and noticed some errors in labeling of Figure 7.

Figure 7: the panels appear to be mis-labeled. The legend says C and D are the schematic, but Panel C in the figure shows the images. I think Panel C should be labeled Panel E. (The panels might need to be rearranged.) Also, it would be nice if the zoomed-in regions before compression were shown in the series.

Suggestions to improve the wording of the text:

Line 59: Suggest adding the citation to Fowler, 2013 as a relatively recent review which includes all the many authors that contributed to the discovery that spectrins are cross-linked by short actin filaments to form a lattice in the erythrocyte membrane.

Lines 44-45: "...several PM binding domains are present along with both α and β subunits. These bonds are the key elements for anchoring..."

Should read: "...several PM binding domains are present along both the α and β subunits. These binding domains are the key elements for anchoring..."

Line 69: "Spectrin role in supporting the PM..." should read "Spectrin's role in supporting the PM..."

Line 66: the citation "JW et al, 2015" should be fixed.

Line 75: the citation "A et al, 2019" should be fixed.

Line 85: I suggest rephrasing: "...while they cooperates during mechanical challenges" to "...but cooperates with cortical actin during mechanical challenges". If the authors do not like this: "they cooperates" should be replaced by "they cooperate".

Line 151: "...but decreased in correspondence of regions of highly positive speeds", is awkward and confusing. Do the authors mean: "...decreased in regions corresponding to highly positive speeds" ?

Line 164: "and were associated to protrusive cell portions" should read: "and were associated with protrusive cell portions"

Line 176: "...drastic remodeling in correspondence with the increased acto-myosin dynamics..." should read "...drastic remodeling corresponding to the increased acto-myosin dynamics..."

Line 196: "spectrin-less zones" would read better as "spectrin-free zones", assuming that there is no spectrin at all, or as "spectrin-depleted zones" as used elsewhere in the manuscript.

Line 208: "focused on cortical regions by different topological organization" would read better as: "focused on cortical regions defined by different topological organization"

Line 413: "...how ubiquitous and evolutionary conserved..." would read better as "...how the ubiquitous and evolutionarily conserved..."

Line 419: suggest replacing "potentially" by "likely"

Line 425-426: Replace "...restricted spectrin binding capability to PM-tangent and optically secluded short actin filaments." with "...restricted spectrin binding capability to a subpopulation of short actin filaments adjacent to the PM". Since the data in this study doesn't tell us anything about whether the actin filaments are tangent to the PM or orthogonal or parallel, there is no need to mention that idea. "optically secluded" is also a strange expression, and I assume the authors mean that they are less abundant than the other filaments, or less detectable. One could perhaps add the word "minor" or "less abundant" in front of "subpopulation", if the authors want to emphasize this aspect.

Line 426: Replace "By far," with "So far,"

Line 454: Suggest adding "network" after "macromolecular" so it reads "macromolecular network condensation".

Line 460: Suggest adding "PM" after "curved" so it reads "retracting curved PM regions".

Lines 463-464: Suggest replacing "reacts" with "responds" and in line 464, also replace "reactions" with "responses". This is not a chemical reaction!

Line 472: replace "through the PM" with "along the PM" (otherwise it sounds as if the spectrin is moving through the bilayer from inside the cell to outside the cell!)

Line 490: I also like "menage a 4"! Except doesn't the "a" need an accent grave?

Reviewer #2 (Remarks to the Author):

The authors responded adequately to my comments and thus have no further suggestions. This work stands as an important contribution to the cytoskeleton field and describes the dynamics of the spectrin network under mechanical stress at an unprecedented level.

Reviewer #3 (Remarks to the Author):

These authors have addressed all my concerns. This is a really outstanding work that I cannot wait to

see in print!!!

Reviewer #4 (Remarks to the Author):

In the revised manuscript, Ghisleni et al. have included additional experiments and statistical analyses to strengthen their observations. While I appreciate the authors' effort, it appears that little additional mechanistic clarity has been provided by the revised manuscript on (1) what causes the complementary localization of actin and spectrin, (2) how spectrin is reorganized by actin and actomyosin contraction, and (3) how mechanical perturbations lead to the redistribution of spectrin and actin.

I understand that these questions may be difficult to address with the cell and micropattern systems used in this study due to the complexity within cells. Fortunately, MPA experiments, as well as coarse grained modeling, have been performed in Duan et al. (2018), which demonstrated that spectrin accumulates to areas of shear deformation, corresponding to the negative curvature on the plasma membrane (at the base of membrane protrusions). The authors' observation in this study (spectrin enriched at negative curvature) is consistent with the conclusion in Duan et al. Therefore, it would be helpful if the authors discuss the findings based on biophysical studies by Duan et al. and provide some explanations to their observations in this study.

Although it is difficult to define the direction of the mechanical forces generated by actomyosin contraction in cells, it would be interesting to localize myosin relative to actin and spectrin. This may provide some clues on how actomyosin contraction reorganizes spectrin localization.

REVIEWERS' COMMENTS:

Reviewer #1 (Remarks to the Author):

This revised manuscript further improves and substantiates the authors' novel findings regarding the complementary dynamics and locations of the actin-rich cortex with respect to the spectrin-based membrane skeleton during cell mechanoresponses. All of my general and specific concerns have been addressed with new experiments, new analyses, as well as clarification and improved interpretations. The new experiments using the micropatterned substrates to demonstrate the complementary domains are excellent, as are the new experiments to test the functions of β spectrin domains in endocytosis. The inclusion of statistical analyses has also substantially improved the confidence in the data, and interpretations and conclusions have been clarified. The concept that the spectrin network responds to mechanical forces by condensation rather than by molecular extension or coiling of individual spectrin molecules is novel. Overall, this study presents exciting and novel findings about spectrin-based membrane skeleton interactions and coordination with actin cytoskeleton dynamics during cell spreading and mechanoresponses. In particular, the common belief has been that the spectrin-based membrane skeleton is only important for highly specialized cells such as neurons and erythrocytes, but this study now shows a critical role for the spectrin-based membrane skeleton in coordinating actin dynamics in so-called 'typical' cultured cell types that are studied by many labs. This will need to be accounted for in future studies.

We are pleased the reviewer recognizes the strength of our work, and appreciate our efforts during the revision process to improve the quality of the manuscript.

I have a few minor suggestions for improvements in the text, and noticed some errors in labeling of Figure 7.

Figure 7: the panels appear to be mis-labeled. The legend says C and D are the schematic, but Panel C in the figure shows the images. I think Panel C should be labeled Panel E. (The panels might need to be rearranged.) Also, it would be nice if the zoomed-in regions before compression were shown in the series.

We agree the labels of Fig. 7 are misleading, as also pointed by the editor and her editorial revisions. We now provide a new version of such figure where panels have been rearranged.

Suggestions to improve the wording of the text:

Line 59: Suggest adding the citation to Fowler, 2013 as a relatively recent review which includes all the many authors that contributed to the discovery that spectrins are cross-linked by short actin filaments to form a lattice in the erythrocyte membrane.

Lines 44-45: "...several PM binding domains are present along with both α and β subunits. These bonds are the key elements for anchoring..."

Should read: "...several PM binding domains are present along both the α and β subunits. These binding domains are the key elements for anchoring..."

Line 69: "Spectrin role in supporting the PM..." should read "Spectrin's role in supporting the PM..."

Line 66: the citation "JW et al, 2015" should be fixed.

Line 75: the citation "A et al, 2019" should be fixed.

Line 85: I suggest rephrasing: "...while they cooperates during mechanical challenges" to "...but cooperates with cortical actin during mechanical challenges". If the authors do not like this: "they cooperates" should be replaced by "they cooperate".

Line 151: "...but decreased in correspondence of regions of highly positive speeds", is awkward and confusing. Do the authors mean: "...decreased in regions corresponding to highly positive speeds" ?

Line 164: "and were associated to protrusive cell portions" should read: "and were associated with protrusive cell portions"

Line 176: "...drastic remodeling in correspondence with the increased acto-myosin dynamics..." should read "...drastic remodeling corresponding to the increased acto-myosin dynamics..."

Line 196: "spectrin-less zones" would read better as "spectrin-free zones", assuming that there is no spectrin at all, or as "spectrin-depleted zones" as used elsewhere in the manuscript.

We thank the reviewer for the suggestion. However, we find "spectrin-less zones" more appropriate in this context.

Line 208: "focused on cortical regions by different topological organization" would read better

as: “focused on cortical regions defined by different topological organization”

Line 413: “...how ubiquitous and evolutionary conserved...” would read better as “...how the ubiquitous and evolutionarily conserved...”

Line 419: suggest replacing “potentially” by “likely”

Line 425-426: Replace “...restricted spectrin binding capability to PM-tangent and optically secluded short actin filaments.” with “...restricted spectrin binding capability to a subpopulation of short actin filaments adjacent to the PM”. Since the data in this study doesn’t tell us anything about whether the actin filaments are tangent to the PM or orthogonal or parallel, there is no need to mention that idea. “optically secluded” is also a strange expression, and I assume the authors mean that they are less abundant than the other filaments, or less detectable. One could perhaps add the word “minor” or “less abundant” in front of “subpopulation”, if the authors want to emphasize this aspect.

Line 426: Replace “By far,” with “So far,”

Line 454: Suggest adding “network” after “macromolecular” so it reads “macromolecular network condensation”.

Line 460: Suggest adding “PM” after “curved” so it reads “retracting curved PM regions”.

Lines 463-464: Suggest replacing “reacts” with “responds” and in line 464, also replace “reactions” with “responses”. This is not a chemical reaction!

Line 472: replace “through the PM” with “along the PM” (otherwise it sounds as if the spectrin is moving through the bilayer from inside the cell to outside the cell!)

Line 490: I also like “menage a 4”! Except doesn’t the “a” need an accent grave?

We would like to thank the reviewer and follow all her/his suggestions (except one) to improve the clarity of the text.

Reviewer #2 (Remarks to the Author):

The authors responded adequately to my comments and thus have no further suggestions. This work stands as an important contribution to the cytoskeleton field and describes the dynamics of the spectrin network under mechanical stress at an unprecedented level.

We would like to thank the reviewer for improving the quality of our work. We are pleased she/he finds the work ‘important’.

Reviewer #3 (Remarks to the Author):

These authors have addressed all my concerns. This is a really outstanding work that I cannot wait to see in print!!!

We are pleased with this comment and thank the reviewer for her/his previous suggestions and enthusiasm!

Reviewer #4 (Remarks to the Author):

In the revised manuscript, Ghisleni et al. have included additional experiments and statistical analyses to strengthen their observations. While I appreciate the authors’ effort, it appears that little additional mechanistic clarity has been provided by the revised manuscript on (1) what causes the complementary localization of actin and spectrin, (2) how spectrin is reorganized by actin and actomyosin contraction, and (3) how mechanical perturbations lead to the redistribution of spectrin and actin.

I understand that these questions may be difficult to address with the cell and micropattern systems used in this study due to the complexity within cells. Fortunately, MPA experiments, as well as coarse grained modeling, have been performed in Duan et al. (2018), which demonstrated that spectrin accumulates to areas of shear deformation, corresponding to the negative curvature on the plasma membrane (at the base of membrane protrusions). The authors’ observation in this study (spectrin enriched at negative curvature) is consistent with the conclusion in Duan et al. Therefore, it would be helpful if the authors discuss the findings based on biophysical studies by Duan et al. and provide some explanations to their observations in this study.

We thank the reviewer for this in depth analysis of strengths and weaknesses of our work. Indeed, we believe our work at single cell level is consistent and complementary with the work by Duan et al, which focus on actin/spectrin mechanoresponse during cell-cell fusion. Our cell compression experiment captured the events that defines the distinct domains of accumulation between acto-myosin and spectrin; indeed the spectrin clearance upon mechanical perturbation was previously missing in the report by Duan et al.

In the discussion we now add the sentence “this observation frames the event that anticipate the distinct spectrin and actomyosin accumulation during shear deformation described by Duan et al.”

Although it is difficult to define the direction of the mechanical forces generated by actomyosin contraction in cells, it would be interesting to localize myosin relative to actin and spectrin. This may provide some clues on how actomyosin contraction reorganizes spectrin localization.

The experiment shown in figure 4F (and related supplementary video 4) aims at describing actomyosin (specifically Myosin Light Chain) and spectrin behavior during blebbistatin washout. We concluded that actomyosin contractility is transmitted at mesoscale range to the spectrin meshwork. We believe our work already provides a starting point for better addressing these molecular mechanisms in the future, along with the specific myosin isoform responsible for spectrin reorganization.